# Late Holocene intensification of the westerly winds at the subantarctic Auckland Islands (51° S), New Zealand

Imogen M. Browne*[1], Christopher M. Moy[1], Christina R. Riesselman[1,2], Helen L. Neil[3], Lorelei G. Curtin**[1], Andrew R. Gorman[1], Gary S. Wilson[1,2,4]

[1]Department of Geology, University of Otago, Dunedin, 9016, New Zealand
[2]Department of Marine Science, University of Otago, Dunedin, 9016, New Zealand
[3]National Institute of Water and Atmospheric Research (NIWA), Wellington, 6021, New Zealand
[4]New Zealand Antarctic Research Institute (NZARI), Christchurch, 8053, New Zealand

*Current address: College of Marine Science, University of South Florida, St. Petersburg FL, 33701, USA
**Current address: Lamont-Doherty Earth Observatory, Columbia University, Palisades NY, 10964, USA

*Correspondence to*: Imogen M. Browne (imogenbrowne@mail.usf.edu)

**Abstract.**

The Southern Hemisphere westerly winds (SHWW) play a major role in controlling wind-driven upwelling of Circumpolar Deep Water (CDW) and outgassing of $CO_2$ in the Southern Ocean, on interannual to glacial-interglacial timescales. Despite their significance in the global carbon cycle, our understanding of millennial- and centennial-scale changes in the strength and latitudinal position of the westerlies during the Holocene (especially since 5,000 yr BP) is limited by a scarcity of paleoclimate records from comparable latitudes. Here, we reconstruct middle to late Holocene SHWW variability using a fjord sediment core collected from the subantarctic Auckland Islands (51° S, 166° E), located in the modern centre of the westerly wind belt. Changes in drainage basin response to variability in the strength of the SHWW at this latitude are interpreted from downcore variations in magnetic susceptibility (MS) and bulk organic $\delta^{13}C$ and atomic C/N, which monitor influxes of lithogenous and terrestrial vs marine organic matter, respectively. The fjord water column response to SHWW variability is evaluated using benthic foraminifer $\delta^{18}O$ and $\delta^{13}C$, both of which are influenced by the isotopic composition of shelf water masses entering the fjord. Using these data, we provide marine and terrestrial-based evidence for increased wind strength from ~1,600-900 yr BP at subantarctic latitudes that is broadly consistent with previous studies of climate-driven vegetation change at the Auckland Islands. Comparison with a SHWW reconstruction using similar proxies from Fiordland suggests a northward migration of the SHWW over New Zealand during the first half of the last millennium. Comparison with paleoclimate and paleoceanographic records from southern South America and West Antarctica indicates a late Holocene strengthening of the SHWW after ~1,600 yr BP that appears to be broadly symmetrical across the Pacific basin. Contemporaneous increases in SHWW at localities on either side of the Pacific in the late Holocene are likely controlled atmospheric teleconnections between the low and high latitudes, and by variability in the Southern Annular Mode and El Niño Southern Oscillation.

## 1 Introduction

The Southern Hemisphere Westerly Winds (SHWW) influence mid-latitude climate and global carbon cycling on a variety of timescales. The position, strength, and symmetry of the SHWW determines precipitation patterns over mid-latitude southern landmasses, and modifies upwelling of nutrient- and carbon-rich water along oceanic fronts associated with the Antarctic Circumpolar Current (ACC; Carter et al., 2009; Dinniman et al., 2012; Lovenduski and Gruber, 2005; Menviel et al., 2008).

Southern Ocean upwelling releases $CO_2$ into the atmosphere and provides nutrients to lower latitudes via northward advection of

Subantarctic Mode Water (SAMW), which forms at the Subantarctic Front (SAF), on the northern margin of the ACC (Carter et al., 2009; Palter et al., 2010; Sarmiento et al., 2004). Westerly wind-driven upwelling therefore plays a major role in modulating Pacific Ocean primary productivity and Southern Ocean air-sea $CO_2$ exchange, on decadal to orbital timescales (Anderson et al., 2009; Lovenduski et al., 2008; Sigman et al., 2010; Skinner et al., 2010; Toggweiler et al., 2006).

Observations and models demonstrate that the Subantarctic Zone (SAZ), between the Subtropical Front (STF) and SAF is the largest sink for atmospheric $CO_2$ in the Southern Ocean (Lenton et al., 2013; McNeil et al., 2007; Metzl et al., 1999, 2006; Takahashi et al., 2002, 2012). Although the STF southwest of New Zealand shifted rapidly south towards the subantarctic during the deglaciation in response to an intensification/southward shift in SHWW, the position of this front has remained relatively stable over the Holocene (Bostock et al., 2015), likely due to topographic steering at the 500 m depth contour (Smith et al., 2013). The

efficiency of the SAZ carbon sink is sensitive not only to latitudinal migrations of winds and associated fronts on glacial-interglacial timescales, but also to decadal changes in atmospheric circulation patterns, particularly, the symmetry of low-level zonal winds (Le Quéré et al., 2007; Metzl 2009; Takahashi et al., 2009; Landschützer et al., 2015). A pronounced southward shift and intensification of the SHWW over the past few decades is associated with a shift towards the positive phase of the Southern Annular Mode (SAM), the dominant mode of internal climate variability in the high southern latitudes, and is thought to be impacting the

efficiency of the SAZ $CO_2$ sink (Abram et al., 2014; Archer and Caldeira, 2008; Landscühtzer et al., 2015; Lovenduski et al., 2008; Marshall, 2003; Shindell and Schmidt, 2004; Thompson and Solomon, 2002). It is therefore important to constrain the zonal symmetry of the SHWW on a variety of timescales in order to understand their influence on air-sea $CO_2$ flux.

Although SHWW circulation is well constrained on seasonal and glacial-interglacial timescales, its behaviour on intermediate

timescales remains unclear. Due to an absence of extensive continental landmasses in the Southern Hemisphere mid-latitudes, atmospheric circulation is zonally symmetric on monthly and longer time scales (Sen Gupta and England, 2006; Hall and Visbeck, 2002; Trenberth, 1991), although internal modes of climate variability such as the SAM and the El Niño Southern Oscillation (ENSO) can influence SHWW symmetry as well as regional sea surface temperatures (SSTs; Ummenhofer et al., 2009; Ummenhofer and England, 2007). Climate data reanalysis studies demonstrate that the SHWW migrate southward in the austral

summer and northward in the austral winter, with a pronounced split jet over the Pacific Basin (Chiang et al., 2014; Garreaud et al., 2013; Trenberth, 1991). A similar pattern is reconstructed on glacial-interglacial timescales, where the westerlies are shifted southward during interglacials and northward during glacials (Anderson et al., 2009; Sigman et al., 2010; Skinner et al., 2010; Toggweiler et al., 2006). Paleoclimate records indicate zonally symmetric winds from 14,000-5,000 yr BP, while a general paucity of records from comparable latitudes across the Pacific and disagreement in interpretations of Holocene SHWW variability have

precluded assessment of zonal symmetry over the past 5,000 years (Fletcher and Moreno, 2012; Kilian and Lamy, 2012; Lamy et al., 2010; Moreno et al., 2009).

In order to assess zonal symmetry of the winds over the middle to late Holocene, a number of well-dated and contemporaneous records of wind strength from areas influenced by the modern SHWW "core" (50-55° S), and from a range of longitudes are

required. Paleoclimate records of wind strength reconstruct changes in atmospheric circulation using the strong positive correlation between wind strength and precipitation in the middle latitudes of the Southern Hemisphere (Garreaud, 2007). Although there are a number of records from southern New Zealand (Gellatly et al., 1988; Knudson et al., 2011; Lorrey et al., 2008, 2014; Putnam et al., 2010; Turney et al., 2017), Tasmania (Saunders et al., 2012), southern South America (Aracena et al., 2015; Ariztegui et al.,

2010; Lamy et al., 2001, 2010; Moreno et al., 2009, 2014; Moy et al., 2008, 2009; Schmipf et al., 2011; Turney et al., 2015; Waldmann et al., 2010 and others), there are fewer from the subantarctic islands, all of which have a paleovegetation focus (McFadgen and Yaldwyn, 1984; McGlone 2002; McGlone et al., 2000, 2010; McGlone and Moar, 1997; Turney et al., 2016).

Fjords along the eastern side of the subantarctic Auckland Islands of New Zealand (51° 40'S, 166° 10'E) contain sedimentary sequences that have continuously accumulated since the end of the last glacial period, representing a unique opportunity to reconstruct post-glacial SHWW variability in the southwest Pacific. Moreover, the physical properties of the water column and the geochemical nature of fjord sediments and benthic foraminifera are influenced indirectly by the overlying atmospheric circulation via the input of wind-driven precipitation, associated erosion in the catchment, and wind-driven mixing. Here, we use a combination of terrestrial and marine proxies from a sediment core recovered from Hanfield Inlet (Figs. 1 and 2) to reconstruct middle to late Holocene SHWW variability and associated changes in shelf hydrography. Drainage basin and fjord circulation responses to variability in the strength and/or position of the SHWW are reconstructed from bulk sedimentary organic carbon isotopes and atomic C/N, which are proxies for organic matter provenance and surface water productivity. In addition, benthic foraminifer stable oxygen and carbon isotopes respond to a combination of bottom water temperature, salinity, productivity, and the input of shelf waters, all of which are driven by the overlying atmospheric circulation.

To accurately reconstruct past changes in atmospheric circulation at the Auckland Islands, modern process studies are required to understand how westerly winds and associated precipitation modify fjord circulation, influence the delivery of organic matter (OM) from both terrestrial ($OM_{terr}$) and marine ($OM_{mar}$) sources, and alter the physical and chemical nature of the water column. In temperate fjords, particularly in the Fiordland region of New Zealand (~500 km to the north of the Auckland Islands), increased precipitation delivers freshwater with abundant terrestrial organic matter to the head and margins of fjords, promoting the formation of a buoyant, seaward-flowing low salinity surface layer enriched in $OM_{terr}$, which is replaced by return flow from the open ocean of saline water enriched in $OM_{mar}$ (Hinojosa et al., 2014; Knudson et al., 2011; Pickrill, 1987; Stanton, 1984. In these settings, the thickness and extent of the low salinity layer primarily depends upon precipitation amount, the size of the catchment relative to the volume of the fjord, the nature of the fluvial drainage network discharging at the head and other locations within the fjord, and wind-induced mixing (Gibbs et al., 2000). Return flow of ocean water is limited by sill height and mixing processes with the overlying low salinity layer (Pickrill, 1987; Stanton, 1984).

With the primary goal of improving understanding of subantarctic Holocene SHWW variability, we address the following research questions: 1) What are the modern circulation processes occurring in the Auckland Islands fjords and how is hydrography related to the wind/precipitation regime? 2) What are the departures of foraminiferal calcite from equilibrium isotopic composition and which species are ultimately the most reliable indicators of bottom water conditions in the fjord? 3) Is there multi-proxy evidence for changing fjord circulation and hydrography that can be linked to concomitant changes in atmospheric circulation over the past 5,000 years? Finally, we compare our Auckland Island record of SHWW variability to contemporaneous records from mainland New Zealand, from similar latitudes (50-53 °S) in southern South America, and to records from West Antarctica, in order to assess symmetry of the westerly wind field across the Pacific basin since the middle Holocene.

## 2 Study Area

The uninhabited subantarctic Auckland Islands are located on the Campbell Plateau, the mostly submarine southeastern extension of New Zealand's continental landmass, and surficial units are composed almost exclusively of late Cenozoic volcanic rock (Adams, 1983). The western coastline is characterised by steep cliffs that are subject to persistent westerly winds and strong ocean currents (Fig. 1). In contrast, the east coast is dominated by sloping ridges, heavily vegetated valleys, inlets and sheltered fjords. The presence of U-shaped valleys, fjords, and other glacial landforms (Fleming et al., 1976), combined with radiocarbon-dated peat sequences immediately overlying glacial sediments indicates that the east coast of the Auckland Islands was likely glaciated during Marine Isotope Stage 2, and that deglaciation was well underway by 15,000 yr BP (McGlone, 2002; McGlone et al., 2000; McGlone and Moar, 1997).

The Auckland Islands are situated at 51° S, within the modern core of the SHWW in the southwest Pacific Ocean (Fig. 1). Like many localities in the southern mid-latitudes, time-averaged zonal SHWW speed is positively correlated with precipitation in the Auckland Islands (Garreaud, 2007), indicating that changes in precipitation through time can be related to past changes in wind strength at this location. At the Auckland Islands, frequent rainfall is combined with persistent cloudiness, gales, and high relative humidity. Meteorological observations since 1991 (Cliflo Database, National Institute for Water and Atmospheric Research) demonstrate little seasonality in rainfall, wind speed and temperature (Fig. S1). Gusts are sourced predominately from the west, exceeding 24 knots for over 300 days of the year, and 33 knots for 200 days of the year. East-west aligned fjords act as funnels to channel the prevailing westerlies, and increased wind stress on the water surface can quickly change fjord circulation and hydrography. Hanfield Inlet is isolated from the open ocean by an entrance sill ~8 m deep, and contains two sub-basins that are ~30 and ~40 m deep (Fig. 2). Tidal range is <1 m at this location. Sediment cores recovered from basins that are protected from the open ocean or shelf currents by bathymetric sills capture changes in lithogenic input and terrestrial vs marine organic matter ($OM_{terr}$ and $OM_{mar}$, respectively), which is related to precipitation-driven runoff.

The Auckland Islands are located in the SAZ of the Southern Ocean, with the STF to the north and the SAF to the south. Shelf waters surrounding the Auckland Islands are cold, fresh, and nutrient-poor (Subantarctic Surface Water; SASW), with SAMW at depths >300 m, sourced from mixing and upwelling along fronts associated with the ACC (Carter et al., 2009). Because of the proximity of the Auckland Islands to these upwelling fronts, changes in westerly-wind driven upwelling will affect the physical and chemical properties of shelf waters entering Auckland Islands fjords that can potentially be recorded by benthic foraminifera. For example, we anticipate that an increase in Ekman-driven upwelling of fresh and nutrient-rich water along the SAF, associated with positive SAM will decrease surface water $\delta^{18}O$ (decreased salinity) and will decrease $\delta^{13}C_{DIC}$ (increased nutrient content).

## 3 Methods

### 3.1 Field Methods

In order to 1) characterise the modern fjord environment, and 2) reconstruct past changes from sediment cores, we collected hydrographic data (conductivity, temperature, depth - CTD) and high-resolution sub-bottom seismic profiles, water samples, surface sediment samples, short box cores, and 6 m long piston cores in Hanfield, Norman, Deep, and McLennan Inlets on the eastern margin of the Auckland Islands in February, 2014 (cruise 14PL001) and 2015 (15PL001), using the University of Otago's R/V *Polaris II* (Table S1; Figs. 1 and 2). Coring sites were selected by identifying locations with two independent sub-bottom seismic profiling systems: a 24-channel hydrophone system with a boomer source (containing frequencies of 100 to 1000 Hz;

potential penetration depth of 200 m below the sea floor) and a higher resolution CHIRP system (containing frequencies of 2-10 kHz; potential penetration depth of 40m below sea floor). A 5.7 m-long piston core (36P4), recovered from 44 m water depth at Site 36 in Hanfield Inlet (Figs. 1d and 2) was sub-sampled for bulk organic stable isotopes ($\delta^{13}$C and $\delta^{15}$N), total organic carbon (% TOC), total inorganic carbon (% TIC) and total nitrogen (% TN), and benthic foraminifer stable isotopes ($\delta^{13}$C and $\delta^{18}$O). An additional 5.5 m-long piston core (39P4), recovered from 52 m water depth at Site 39 in Hanfield Inlet was examined for sedimentological changes. A short (0-10 cm) box-core containing the sediment-water interface from Hanfield Inlet (36B2) and grab samples from Norman (35G1) and Hanfield (36G1 and 39G1) Inlets, with accompanying CTD data were sampled for foraminifer stable isotope analysis ($\delta^{13}$C and $\delta^{18}$O), in order to assess the isotopic departure of foraminiferal calcite from bottom water equilibrium. In addition, 36B2 was sub-sampled every cm for 10 cm and each interval stained with Rose Bengal solution (2 gL$^{-1}$ in ethanol) in order to assess the depth habitat of benthic foraminifera species. Foraminifer tests that are pink (more than 50% of the test) are regarded to be living, or recently dead at the time of collection.

A series of water column samples from fjords and the continental shelf were collected and filtered, and both water and particulates were analysed for stable isotopes ($\delta^{13}$C$_{DIC}$/ $\delta^{18}$O and $\delta^{13}$C/ $\delta^{15}$N respectively). Fjord waters were collected in 10 L Niskin bottles and transferred to 60 mL glass serum bottles for $\delta^{13}$C of dissolved inorganic carbon ($\delta^{13}$C$_{DIC}$) and 30 mL glass serum bottles for $\delta^{18}$O. Samples for $\delta^{13}$C$_{DIC}$ were immediately poisoned with a supersaturated HgCl$_2$ solution to stop biological activity. Water was also collected in 5 L bottles for particulate organic carbon and nitrogen concentrations, and stable isotope composition. These samples were filtered onboard through pre-combusted Whatman GF/C glass fibre filters using a vacuum pump manifold, acidified with 10 % hydrochloric acid to remove any carbonate phases, and oven-dried at 40° C.

## 3.2 Laboratory Methods

### 3.2.1 Water Column Stable Isotopes

Seawater samples collected in 2014 were analysed for oxygen isotopes using a Picarro 2120 wave-length-scanned cavity ring-down spectrometer (WS-CRDS), in the Isotrace Laboratory at the University of Otago. The average standard deviation for 11 duplicate measurements (fjord, lake, and stream samples) was 0.03 ‰ for $\delta^{18}$O. The stable isotopic compositions of dissolved inorganic carbon ($\delta^{13}$C$_{DIC}$) in 2014 samples, and $\delta^{18}$O for the 2015 samples were measured using a Thermo Delta Plus Advantage isotope ratio mass spectrometer (IRMS) interfaced to a Gasbench II via a GC PAL autosampler at the Isotrace Laboratory at the University of Otago. The average standard deviation for 11 duplicates (fjord, lake, and stream samples) was 0.11 ‰ for $\delta^{13}$C$_{DIC}$ and 0.02 ‰ for $\delta^{18}$O (11 duplicates; all fjord samples). Stable nitrogen and carbon isotopes of filtered particulates collected in 2014 were measured using a Carlo Erba NC2500 Elemental Analyser, coupled with a Europa Scientific continuous-flow Isotope Ratio Mass Spectrometer (IRMS) using USGS-40 and USGS-41 standards, in the Isotrace Laboratory at the University of Otago. Average standard deviation for USGS-40 was 0.10 ‰ for $\delta^{15}$N and 0.04 ‰ for $\delta^{13}$C, and for USGS-41 was 0.20 ‰ for $\delta^{15}$N and 0.05 ‰ for $\delta^{13}$C. Average standard deviation for duplicate filtered particulates was 1.8 ‰ for $\delta^{15}$N (n=4) and 0.25 ‰ for $\delta^{13}$C (n=7). Average standard deviation for duplicate bulk sediment was 0.22 ‰ for $\delta^{15}$N (n=9) and 0.06 ‰ for $\delta^{13}$C (n=9). Stable nitrogen and carbon isotopes of filtered particulates collected in 2015 were measured using a Carlo Erba NA1500 Series 2 Elemental Analyzer coupled to a Thermo Finnigan Delta Plus via a ConFlo II open split interface at the Stanford University Stable Isotope Lab (SIBL). Average standard deviation for duplicate filtered particulate samples was 2.73 ‰ for $\delta^{15}$N, 0.24 ‰ for $\delta^{13}$C and 1.2 for atomic C/N (n=1 pair for $\delta^{15}$N and n=2 pairs for $\delta^{13}$C and C/N). Average standard deviation for USGS-40 was 0.13 ‰ for $\delta^{15}$N and 0.09 ‰ for $\delta^{13}$C (n=27).

### 3.2.2 Bulk Organic Stable Isotopes

Surface and downcore sediment samples collected in 2014 were freeze-dried, homogenised, and weighed into tin and silver capsules. Increasing volumes of 6 % sulphurous acid were added to silver capsules to remove carbonate phases. Unacidified samples were weighed into tin capsules for concentration and isotopes of organic nitrogen (% TN and $\delta^{15}$N). Both acidified (silver) and unacidified (tin) samples were measured for concentration and isotopes of organic carbon (% TOC and $\delta^{13}$C) using a Carlo Erba NC2500 Elemental Analyser, coupled with a Europa Scientific continuous-flow IRMS at the Isotrace laboratory at the University of Otago. Average standard deviation for nine duplicate samples was 0.22 ‰ for $\delta^{15}$N and 0.06 ‰ for $\delta^{13}$C. The abundance of biogenic carbonate (% TIC) was estimated by subtracting the organic carbon concentrations (silver capsule) from total carbon concentrations (tin capsules).

### 3.2.3 Radiocarbon Analyses and Chronology

A chronology for core 36P4 was developed using eight AMS radiocarbon-dated acid-base insoluble terrestrial organic macrofossil fragments derived mostly from the >500 μm size fraction (Table 1; Fig. 3). Samples were analysed at the Center for Accelerator Mass Spectrometry (CAMS) at Lawrence Livermore National Laboratory, California, USA. Radiocarbon ages were calibrated using Calib 7.0.4 (Stuvier and Reimer, 1993) and a Southern Hemisphere atmospheric calibration curve (SHCal13; Hogg et al., 2013). Two samples collected 90 cm apart near the base of the core returned median probability ages that were approximately the same, indicating rapid sediment deposition and/or sediment reworking between 370 and 450 cm. We therefore focus our paleoclimate record on the interval from 0-420 cm (approximately 4,000 yr BP), below which smear slides begin to contain fragmented sponge spicules, and where magnetic susceptibility abruptly increases to the highest values observed in the core. A Bayesian modelling approach was used to model the distribution of time throughout the core stratigraphy, and the oldest two [14]C dates from the disturbed section were not included in this model. Bacon 2.2 (Blaauw and Christen, 2011) was used, with the following priors: accumulation shape = 1.5, accumulation mean = 10 yrcm[-1], memory mean = 0.7, strength mean = 4 (Z), with a 71.5 cm section. The weighted mean calibrated age BP from the 95 % confidence interval of the Markov Chain Monte Carlo simulations is used. The age model indicates that the core top is approximately 400 years old, indicating that the most recent part of the record was not recovered during piston coring.

### 3.2.4 Benthic Foraminifer Stable Isotopes

Benthic foraminifera *Cibicides* spp., *Nonionellina flemingi, Trifarina angulosa,* and *Bulimina marginata* f. *marginata* were picked from the 125-250 μm fraction, and *Quinqueloculina seminula* from the <250 μm fraction, and were cleaned by briefly sonificating in distilled water. Foraminifer $\delta^{13}$C and $\delta^{18}$O were measured using a Kiel IV carbonate device, coupled to a MAT-253 Mass Spectrometer at the National Institute of Water and Atmospheric Research (NIWA) in Wellington, New Zealand. Internal precision of the instrument was monitored by measuring NBS-19 with each set of unknown samples and standard deviation for NBS-19 was 0.01-0.03 ‰ for $\delta^{13}$C and 0.02-0.07 ‰ for $\delta^{18}$O. The standard deviation of one replicate sample for *Cibicides* spp. (from 36P4) is 0.10 ‰ for both $\delta^{13}$C and $\delta^{18}$O. The average standard deviation of replicate samples for *N. flemingi* (from 36P4) was 0.27 ‰ for $\delta^{13}$C (n=3) and 0.07 ‰ for $\delta^{18}$O (n=4).

### 3.2.5 Calculating Vital Offset of Benthic Foraminifer $\delta^{18}$O from Equilibrium

Modern bottom water $\delta^{13}$C$_{DIC}$, $\delta^{18}$O ($\delta^{18}$O$_w$) and temperature were measured at Site 35 (Norman Inlet; CTD_001), and Sites 36 and 39 (Hanfield Inlet CTD_002 and CTD_004, Table S2). Expected foraminifer oxygen isotopic composition precipitated in equilibrium with bottom water ($\delta^{18}$O$_c$) for a given temperature (T) was calculated for each site using Eq.(1), a linear

paleotemperature equation derived by Marchitto et al., (2014) (Table 2). Equation 1 is indistinguishable from that of Kim and O'Neil (1997), derived from experiments involving precipitation of inorganic calcite in seawater at different temperatures hence

is here considered appropriate for species other than *Cibicides*.

$$(\delta^{18}Oc - \delta^{18}Ow(VSMOW) + 0.27) = -0.225T + 3.50$$
(1)

**Equation 1. Where δ$^{18}$O$_c$ is average oxygen isotopic composition of calcite precipitated in equilibrium with bottom water δ$^{18}$O, δ$^{18}$O$_w$ is measured bottom water oxygen isotopic composition (relative to VSMOW), and T is bottom water temperature in °C (Marchitto et al.,**
**2014). A correction factor of 0.27 ‰ is applied when converting δ$^{18}$O from the VSMOW to VPDB scale (Hut 1987).**

## 4 Results

### 4.1 Seismic profiles

The images reveal packages of stratified sediment >10 m thick, free of sediment gravity flows (Fig. 2). The basin immediately landward of the entrance sill in Hanfield Inlet is interpreted to contain two distinctive layers of sediment 4-6 m and ~2 m thick,
respectively, separated by a prominent seismic reflector. A transition to units with higher seismic velocity occurs beneath the shallow sedimentary layers.

### 4.2 Modern Fjord Hydrography and Water Column Stable Isotopes

CTD profiles and water isotopes measured landward of submarine sills during the 2014 and 2015 field seasons reveal that the fjords are only stratified for very short intervals of time (Fig. 4). During 1-2 day intervals when high-pressure is situated over the
Auckland Islands, causing calm winds, no precipitation, relatively warm air temperatures, and limited cloud cover, surface water temperature (upper ~4 m) and δ$^{18}$O (upper ~10 m) are elevated, and salinity is slightly depressed (upper ~4 m) (eg. CTD_007; Fig. 4a). During intervals when a low-pressure system is passing over or is in the vicinity of the Auckland Islands, precipitation and high wind velocities persist, and temperature and salinity show little variation with depth (eg. CTD_006; Fig. 4b). Particulate δ$^{13}$C and δ$^{15}$N are somewhat more positive (+ ~1 ‰) during rainy and high wind synoptic conditions (Fig. 4b) compared to times when
dry / low wind velocities persist (Fig. 4a), although there is no significant difference in the vertical isotopic gradient in either regime. Atomic C/N of particulate organic matter (OM) does not vary throughout the water column during rainy / high wind conditions, but appears to be elevated in the subsurface during dry / low wind conditions (Fig. 4a). Physical properties and water isotopes measured in 2014 on the continental shelf between Hanfield and Norman Inlet demonstrate little variation in temperature, salinity, δ$^{18}$O, and particulate δ$^{15}$N and δ$^{13}$C with depth (Table S2). Average salinity outside of the fjords is 34.5, temperature is
10.5° C, δ$^{18}$O is -0.14 ‰ and δ$^{13}$C$_{DIC}$ is 0.15 ‰. Fjord salinity during sunny and dry conditions is similar to outside of the fjord, but is depressed by ~0.25 during rainy and high wind synoptic conditions (Fig. 4).

Filtered particulate samples are separated into five groups based on C and N isotopic composition: low salinity layer (particulate OM samples from depths <10 m in Hanfield and Norman Inlets), deep fjord (particulate OM samples from depths >10 m in
Hanfield and Norman Inlets) offshore (particulate OM samples from the continental shelf seaward of the sill), lake samples (particulate OM samples from 3 lakes on the main island), and fjord surface sediment samples from Hanfield Inlet (Fig. S2; Table S2). There is no obvious difference between fjord surface water and deep water stable isotopic ratios and atomic C/N. The δ$^{13}$C of fjord particulate OM samples generally exhibits more positive values than the shelf surface waters, (offshore) although CTD_003 taken from outside Norman and Hanfield Inlets has a more positive average δ$^{13}$C composition (-21.52 ‰) and lower average C/N
(5.51) relative to Hanfield Inlet during dry and low wind conditions (-23.1 ‰ and 7.17, respectively). Samples collected farther offshore from Chambres Inlet and near Stewart Island in 2014 have δ$^{13}$C values that average -22.01 ‰. Sediment samples from

Hanfield Inlet exhibit an intermediate $\delta^{13}C$ value and C/N is elevated by ~2-3 units relative to particulate OM samples. There are no obvious patterns in $\delta^{15}N$ across different sample groups.

## 4.3 Modern Benthic Foraminifer Stable Isotopes

Averaged $\delta^{18}O_{foram}$ for selected species of benthic foraminifera from grab and box-core samples in both Norman and Hanfield Inlets are compared to $\delta^{18}O_c$, calculated from measured $\delta^{18}O_w$ and temperature during the summers of 2014 and 2015 (Tables 2 and 3, Fig. 5). Epifaunal *Cibicides* spp. (n=3) precipitates $\delta^{18}O$ within the range of expected $\delta^{18}O_c$, hence in equilibrium with bottom water. Shallow infaunal New Zealand endemic *N. flemingi* (n=5) is enriched in $^{18}O$ relative to what is predicted for equilibrium fractionation. Other species including *B. marginata* f. *marginata* (n=5)*, T. angulosa* (n=6) and *Q. seminula* (n=6) also 
show positive offsets relative to $\delta^{18}O_c$, and the degree of offset generally increases with increased depth habitat. Rose Bengal staining of box-core samples indicates that *B. marginata* f. *marginata* and *C. lobatulus* live at 0-1 cm depth, *N. flemingi, Sigmoilopsis elliptica* and *Anomalinoides sphericus* at 1-2 cm, *Cassidulina carinata* at 2-3 cm and *Bolivina* cf. *earlandi* at 3-4 cm. Although *T. angulosa* was not found in this study, it is known to be shallow infaunal, and lives at 0-1cm depth (Mackensen et al., 1990). *Quinqueloculina. seminula* was likewise unstained, so the depth habitat in Hanfield remains unconstrained. However, Scott 
et al., (2001) indicate that this species is shallow infaunal or possibly epifaunal.

## 4.4 Sedimentology, Magnetic Susceptibility, and Bulk Organic Carbon and Nitrogen Isotopes

Sediment core 36P4 is composed of homogenous dark brown fine-grained sand and mud, faecal pellets, small (<500 μm) benthic and planktonic foraminifera, gastropods, ostracods, calcareous nannofossils, diatoms, and terrestrial plant debris. The upper ~90 cm of the core is slightly bioturbated and there are no significant lithological distinctions (colour, grainsize, sediment type) 
downcore. Sediment core 39P4 contains a sharp transition from brown foraminifer-bearing fine-grained sand and mud to dark freshwater diatom-bearing clays.

Magnetic susceptibility (MS), wt % TN, TOC and TIC, bulk organic $\delta^{13}C$ and $\delta^{15}N$, and atomic C/N are plotted against depth in Fig. 6 and age in Fig. 7. From ~4,800-4,000 yr BP, MS is highly variable, wt% TN, TOC and TIC is relatively low. From ~4,000- 
1,600 yr BP, MS, wt % TN, TOC and TIC and bulk organic $\delta^{13}C$ and C/N remain relatively unchanged. From ~1,600-500 yr BP, wt % TN and TOC increases, bulk organic $\delta^{13}C$ is more positive, $\delta^{15}N$ is more negative, and atomic C/N becomes lower.

## 4.5 Downcore Benthic Foraminifer Stable Isotopes

Downcore isotope results for *Cibicides* spp. and *N. flemingi* are shown in Table S3, Fig. 8, and Fig. S3, with no correction for species-specific vital offsets. The chrostratigraphically-constrained portion of the core is subdivided into three sections (Fig. 7), 
based on high amplitude variations (>1‰) in foraminifer $\delta^{13}C$ and $\delta^{18}O$. Using this approach, we observe that other proxies also shift at these boundaries (Section 4.4). Both isotopes of epifaunal *Cibicides* spp. exhibit similar trends downcore, with more negative $\delta^{13}C$ values corresponding to more negative $\delta^{18}O$ (Fig. S3). The range of variability in both isotopes increases above 100 cm depth (~1,600 yr BP). The overall trend and range of $\delta^{18}O$ for both species is similar, whereas the trend and range for $\delta^{13}C$ differs between the two (Fig. 8). *Nonionellina flemingi* exhibits more negative $\delta^{13}C$ values (downcore range from ~0.5 to -3.5 ‰) 
whereas *Cibicides* spp. demonstrates more positive $\delta^{13}C$ values (downcore range from ~0 to -1.5 ‰).

**5 Discussion**

**5.1 Modern Fjord Circulation and Response to SHWW Variability**

CTD data collected from Hanfield and Norman Inlets in 2014 and 2015 indicate the absence of a significant low salinity layer or any persistent stratification of the water column (Fig. 4). Our limited summer observations indicate that although the upper ~4 m of the water column can thermally stratify by 0.7° C, the water column quickly becomes isothermal once the next frontal system passes over the islands. Moreover, small catchment/fjord areas result in limited fluvial discharge at the head of the fjord which prevents formation of a significant low salinity layer, while strong winds orientated parallel to the fjord axis mix the water column

and further prevent the establishment of significant estuarine circulation.

The isotopic composition and atomic C/N of particulate OM can be used to determine sediment provenance in continental margin settings, and when measured downcore in fjord sediment cores, these parameters can be used as a potential proxy for $OM_{terr}$ delivery derived from westerly precipitation (Hinojosa et al., 2014; Knudson et al., 2011; Meyers and Teranes, 2001; Walinksy et al., 2009).

$OM_{terr}$ is characterised by more negative $\delta^{13}C$ and $\delta^{15}N$ whereas $OM_{mar}$ is characterised by more positive $\delta^{13}C$ and $\delta^{15}N$, although it is important to note that isotopes of bulk sediment organic matter can also be influenced by surface water productivity in a stratified water column and potentially by post-depositional degradation by bacteria (Meyers and Teranes, 2001). Similarly, atomic C/N can be helpful to discriminate between terrestrial and marine OM sources (Perdue and Koprivnjak, 2007). Terrestrial vascular plants are cellulose-rich and contain a relatively high proportion of carbon compared to marine algae which are protein-rich and

contain more nitrogen (Meyers and Teranes, 2001). A higher C/N ratio therefore is indicative of a larger contribution of $OM_{terr}$, relative to $OM_{mar}$ and vice versa.

Hanfield Inlet demonstrates a distinct hydrographic response to changes in SHWW-derived precipitation and mixing (Fig. 4). During intervals when prevailing wind speed and precipitation are low, the fjord is weakly stratified due to solar heating (elevated

temperature) and evaporation (more positive $\delta^{18}O$/ increased salinity) in the uppermost part of the water column (Fig. 4a). During periods with high winds and rainfall, this weak stratification rapidly breaks down, and there is no vertical structure in temperature, salinity, and C/N, likely due to wind-induced mixing and subsequent overturning of the water column (Fig. 4b). Also during periods of high winds and precipitation, water column salinity is slightly depressed, relative to periods of low winds and precipitation. We do not observe a reduction in fjord bottom water $\delta^{18}O$ during rainy/high wind conditions, but this is likely due

to our inability to sample the water column across a range of seasons/years. We speculate that sustained freshwater input into the fjord and wind-induced mixing of the water column does reduce the $\delta^{18}O$ of bottom water. Increased rain is also expected to promote erosion and delivery of lithogenic material into the fjord basin, although this is not assessed in this study.

Bulk sedimentary stable isotopes and atomic C/N ratios provide further insight into the hydrographic response of fjords to increased

westerly wind-derived precipitation (Fig. 4; Fig. S2; Table S2). In contrast to fjords in Fiordland, southern New Zealand, Auckland Islands particulate data indicate a greater contribution of $OM_{mar}$ (more positive $\delta^{13}C$ and lower C/N during periods of sustained high winds, suggesting that an increased delivery of $OM_{mar}$ through entrainment in the return flow of shelf waters overwhelms the signal of increased freshwater input from runoff (with entrained $OM_{terr}$). Because the Auckland Islands fjords are shallower and shorter, have much smaller catchment areas and experience lower precipitation than those of southern New Zealand, a significant

low salinity surface layer is not created. Nevertheless, because bulk sedimentary $\delta^{13}C$ is more positive and C/N is lower during windy periods relative to low wind conditions, these parameters can be used as indirect proxies for wind strength at the Auckland Islands. Because bulk sedimentary $\delta^{15}N$, which can be influenced by additional processes such as denitrification and substrate

evolution (Altabet and François, 1994; Meyers and Teranes, 2001), shows no discernible difference in isotopic composition between water masses inside and outside of the fjord (Fig. S2), it is not incorporated into our paleoclimate discussion.

In summary, our observations indicate that an increase in wind strength and associated precipitation at the Auckland Islands induces complete mixing of the fjord water column, reduces salinity, and increases the contribution of $OM_{mar}$. We speculate that increased wind strength induces complete mixing of fjord waters, and that wind stress aligned parallel to the axis of east-west orientated fjords increases the outflow of surface waters, driving return flow of $OM_{mar}$-dominated shelf waters. Further hydrographic studies of Auckland Island fjords during different seasons would be required to vigorously test this hypothesis, however given present

information and our hypothesised link between westerly wind-driven precipitation and delivery of OM into the fjord, downcore changes in bulk sedimentary $\delta^{13}C$ and C/N are interpreted here as indirect proxies for wind strength, with more positive $\delta^{13}C$ and a lower C/N indicative of high winds and associated precipitation.

**5.2 Modern Benthic Foraminifer Oxygen Isotopes**

Epifaunal *Cibicides* spp. recovered from box-core and grab samples in Hanfield Inlet precipitates within the range of calculated

$\delta^{18}O_c$ for the entire range of temperatures (9.9-10.55 °C) and $\delta^{18}O_w$ (-0.26 to 0.31‰) measured in Hanfield and Norman Inlets (Fig. 5). Previous studies have also demonstrated that *Cibicides* spp. precipitates in equilibrium with ambient bottom waters (Graham et al., 1981; Grossman, 1984; Hald and Vorren, 1987; Lynch-Stieglitz et al., 1999; Marchitto et al., 2014; McCorkle et al., 1997). New Zealand endemic and infaunal *N. flemingi* also recovered in box-core and grab samples from Hanfield and Norman Inlets exhibits isotopic values that are more positive relative to $\delta^{18}O_c$, similar to the Northern Hemisphere species *N. laboradorica*

(Grossman, 1984).

**5.3 Middle to Late Holocene SHWW Variability**

Geophysical and sedimentological data from Hanfield Inlet reveal the deglacial and post-glacial history of Hanfield Inlet. A transition in core 39P4 (collected 0.5 km to the east of 36P4; Figs 1 and 2) is characterised by an abrupt change from dark, organic-rich mud with freshwater diatoms to foraminifer-bearing fine-grained sand and mud, indicating that the fjord once contained a lake

landward of the entrance sill. Two sharp transitions identified in seismic profiles occur beneath the fine-grained marine and lacustrine sedimentary sequence and probably represent glacial sediment deposited on eroded volcanic basement during glacial retreat. Based on the height of the submarine sill (~8 m) and relative sea level curves from southern New Zealand (Clement et al., 2016; Dlabola et al., 2015), marine incursion of Hanfield Inlet likely occurred between 10,000 and 9,000 yr BP.

Sedimentological evidence for disturbance and radiocarbon results from 36P4 indicate that the bottom part of the core (370-450 cm) contains reworked sediment, probably the result of a single localised slump event. For this reason, the oldest $^{14}C$ date incorporated into the Bayesian age model for 36P4 was that at 366 cm (4691 ± 30 yr BP), and only the undisturbed section of the core is used for SHWW reconstruction.

Fjord conditions are relatively stable from ~4,000 to 1,600 yr BP (260-115 cm), as indicated by the generally unchanging MS, wt % of TN, TOC and TIC, and bulk organic $\delta^{13}C$ and atomic C/N (Figs. 6 and 7). This entire interval is characterised by more negative bulk sedimentary $\delta^{13}C$ and elevated atomic C/N, compared to the uppermost part of the core <1,600 yr BP (115-0 cm; Fig. 6), indicating a smaller contribution of $OM_{mar}$ and therefore an overall increased stratification and decreased SHWW strength during this interval.


Concomitant shifts in MS and bulk sedimentary data at ~1,600 yr BP (115 cm) and ~900 yr BP (40 cm) indicate increased variability in fjord hydrography related to changes in atmospheric circulation during the late Holocene (Figs. 6 and 7). From 1,600-900 yr BP (115-40 cm), increased wt % TN, more positive bulk organic sedimentary $\delta^{13}$C and lower C/N indicates an increased contribution of OM$_{mar}$ to the fjord, suggesting invigorated wind-driven mixing and return flow of shelf waters. In addition,
increased MS during this interval during which non-lithogenic sedimentary components (wt % TN, TOC and TIC) remain stable indicates increased supply of lithogenic material via runoff, corroborating the interpretation of this period as an interval of high winds/ precipitation. From 900-500 yr BP (40-0 cm), a decrease in MS is accompanied by an increase in % TN and TOC, which could represent dilution of lithogenic material as well as reduced input (Figs. 6 and 7). Increased % TN and TOC at this time probably represent increased surface water productivity (rather than an increase in precipitation and wind strength), which is
reflected by more positive $\delta^{13}$C. Increased fjord water productivity could be related to increased stratification under weakened winds or input of nutrient-rich waters from outside of the fjord. From 1,600-500 yr BP (115-40 cm), bulk sedimentary $\delta^{13}$C and $\delta^{15}$N become decoupled (Fig. 6), but because $\delta^{15}$N in the modern fjord system is unrelated to the source of organic matter (Fig. S2), it is not considered a suitable proxy to interpret changes in westerly wind-driven precipitation in Hanfield Inlet.

Superimposed on long-term paleoenvironmental and paleoclimate changes are sub-millennial scale variations in fjord bottom water chemistry, as indicated by paired changes in benthic foraminifer stable isotopes (Figs. 7 and 8). Covariance of both $\delta^{18}$O and $\delta^{13}$C for epifaunal *Cibicides* spp. (Fig. S3) indicates that fjord hydrography is influenced by the introduction of shelf waters with different physical properties and an isotopically distinct composition (our limited observations indicate that shelf water $\delta^{18}$O and $\delta^{13}$C$_{DIC}$ are characterised by more negative values than fjord waters; Table S2). $\delta^{18}$O of infaunal species *N. flemingi* covaries with
*Cibicides* spp. $\delta^{18}$O, while *N. flemingi* $\delta^{13}$C differs, reflecting the porewater $\delta^{13}$C$_{DIC}$ influence often observed in infaunal $\delta^{13}$C (Mackensen et al., 2000; McCorkle et al., 1990), masking than ambient bottom water conditions. Overall, covariance between infaunal and epifaunal $\delta^{18}$O and disparity in $\delta^{13}$C indicates that there is no significant post-depositional overgrowth in Hanfield Inlet benthic foraminifera (Fig. 8).

The oxygen isotopic composition of benthic foraminifera is dependent on temperature, salinity, species-specific offsets, and the isotopic composition of the source water (Ravelo and Hillaire-Marcel, 2007). Sub-millennial changes in the $\delta^{18}$O downcore stratigraphy are as large as 1.0‰ (Figs. 7 and 8), which would correspond to fjord bottom water temperature changes greater than 4°C (~+0.23‰ per 1°C increase; Marchitto et al., 2014). This is unlikely given regional terrestrial temperature reconstructions (McGlone et al., 2010), and the <0.5 ‰ range of Holocene $\delta^{18}$O variability in planktic foraminifera derived from a transect of
sediment cores collected in the Solander Trough close to the Auckland Islands (Bostock et al., 2015). Therefore, a combination of local salinity effects, temperature changes, and the $\delta^{18}$O of regional waters likely influence the $\delta^{18}$O of benthic foraminifera in Hanfield Inlet. Measured summer salinity profiles indicate a range in salinity from 34.5 to 34, which corresponds to a 0.2 ‰ decrease in $\delta^{18}$O$_c$ (Lynch-Stieglitz et al., 1999), although given our limited measurements (two field seasons), the fjord salinity minimum could be lower during times of sustained input and mixing of westerly wind-derived fresh water into the fjords. Based
on our observations, we interpret more negative benthic foraminifer $\delta^{18}$O to reflect enhanced westerlies, which increases the input and mixing of more negative $\delta^{18}$O freshwater into the fjords, prohibits long-term stratification of the fjord, and allows for the introduction of more southerly sourced Southern Ocean water (with a stable isotopic composition of ~ -0.5 ‰; Boyer et al., 2013), sourced from northwards export of waters upwelled along the SAF. Alternatively, increased SST, which is shown in data reanalysis studies to be related to a positive phase of the SAM, associated with increased westerly strength over this part of New Zealand
(Ummenhofer et al., 2009; Ummenhofer and England, 2007) would cause a decrease in surface water $\delta^{18}$O.

Benthic foraminifer $\delta^{13}C$ reflects ambient bottom water $\delta^{13}C_{DIC}$, which is controlled by air-sea exchange of $CO_2$, biological processes of photosynthesis and respiration, and addition of water masses with signature isotopic characteristics (Ravelo and Hillaire-Marcel, 2007). Increased input of freshwater during periods of intensified winds and rainfall would decrease $\delta^{13}C_{DIC}$

because of increased respiration and remineralisation of $^{12}C$-rich $OM_{terr,}$ which is consistent with our interpretation of decreased $\delta^{18}O$ also being indicative of increased wind strength. In addition, fjord water $\delta^{13}C_{DIC}$ is most likely to be influenced by water masses entering from outside of the fjord. In the SAZ of the Southern Ocean, a model based on satellite data (Lovenduski and Gruber, 2005) demonstrate positive SST anomalies (manifested as more negative foraminifer $\delta^{18}O$ in the current record) and lower surface water productivity (inferred by low chlorophyll concentrations), associated with a positive SAM and intensified SHWW.

Increased water column stratification because of elevated SST is hypothesised to reduce productivity, by limiting macronutrient supply to the euphotic zone (Lovenduski and Gruber, 2005), which will cause $\delta^{13}C_{DIC}$ to be more negative because less of the light isotope, $^{12}C$ can be preferentially utilised by macro nutrient-limited phytoplankton.

Other modelling studies have shown that the recent southward shift in the SHWW has increased upwelling and northward advection

of SAMW at ~60°S (Le Quéré et al. 2007; Lovenduski et al., 2008; Wetzel et al., 2005). Northward export of SAMW with more negative $\delta^{13}C_{DIC}$ (contains more $^{12}C$, released by remineralisation of organic matter at depth) could also contribute to lowering of SAZ surface water $\delta^{13}C_{DIC.}$ Taken together, these results suggest that variations in the $\delta^{18}O$ and $\delta^{13}C$ of benthic foraminifera recorded in Hanfield Inlet are controlled indirectly through the influence of the strength and latitudinal position of the SHWW on shelf/fjord hydrography during the middle to late Holocene. The foraminifer isotope record from Hanfield Inlet exhibits more

negative $\delta^{18}O$ (decreased salinity from local rainfall and fjord mixing and an increased influence of a southerly-sourced water masses), and more negative $\delta^{13}C_{DIC}$ (increased Ekman upwelling and northward export of SAMW), interpreted here to be the result of strengthened SHWW in the southwest Pacific Ocean. This interpretation is consistent with modelled increased SST and decreased surface water productivity in the SAZ under positive SAM (Lovenduski and Gruber, 2005), although it is important to note that extra-tropical variability driven by ENSO also affects SSTs around New Zealand (Ummenhofer et al., 2009). In addition,

increased wind-driven upwelling and northward export of SAMW with depleted $\delta^{13}C_{DIC}$ (Lovenduski et al., 2007; Le Quéré et al., 2007; Wetzel et al., 2005) could contribute to lowering $\delta^{13}C_{DIC}$ at the mid-latitudes.

Our middle to late Holocene paleoclimate reconstruction at the Auckland Islands, using both terrestrial and marine proxies indicates increased influence of the SHWW during the late Holocene. Around 1,600 yr BP (115 cm), bulk sedimentary $\delta^{13}C$ becomes more

positive and atomic C/N decreases, indicating an increased contribution of $OM_{mar}$ as a result of wind-induced mixing and enhanced return flow of shelf waters. Increased variability in benthic foraminifer $\delta^{18}O$ and $\delta^{13}C$ during the late Holocene represents enhanced SHWW influence. Paired decreases in benthic foraminifer $\delta^{18}O$ and $\delta^{13}C$ are interpreted as increased input of shelf waters with characteristic water properties and mixing of freshwater into the fjord, both processes which are related to increased SHWW strength in the southwest Pacific Ocean. From ~900-500 yr BP (40-0 cm), surface productivity increases, possibly related to

increased fjord stratification due to a reduction in westerly wind-derived precipitation, or changes in shelf water hydrography.

### 5.4 Comparison to Other Records of SHWW Variability

### 5.4.1 New Zealand

Previous Holocene SHWW reconstructions at the Auckland and other subantarctic islands are based on palynology from peat cores (McGlone, 2002; McGlone et al., 2000, 2010; McGlone and Moar, 1997) and tree-line reconstructions (Turney et al., 2016), and

are broadly consistent with our interpretations of increased SHWW influence from ~1,600-900 yr BP (Fig. 9). Inconsistencies in the records including periods of enhanced/ reduced SHWW influence in the late Holocene may be due to varying sensitivity of different proxies to westerly wind strength and/ or chronological uncertainty. Tree-line reconstructions suggest increased wind strength from 2,000-1,000 yr BP (Turney et al., 2016), and the establishment of tall *Metrosideros* forest from 5,500-4,000 yr BP (McGlone et al., 2000) suggests strengthened SHWW and decreased cloudiness over the middle and late Holocene. Wide-spread

retreat of *Metrosideros* forest and replacement by woody shrubs and trees such as *Dracophyllum*, *Myrsine,* and *Raukaua* in a bog core after 1,500 yr BP indicates higher water tables and increased exposure (McGlone et al., 2002). McGlone et al. (2000) also demonstrate several millennial-scale expansions and collapses of *Metrosideros* forest since 4,000 yr BP, similar to millennial-scale fluctuations in SHWW variability which is manifested in our foraminifer isotope record of shelf water hydrography.

A comparison of SHWW records from Fiordland (45° S, 166-167° E) with the current record (51° S) allows for assessment of latitudinal shifts in the wind field over New Zealand. Two intervals of elevated westerly wind-derived precipitation in Fiordland reconstructed using bulk organic stable isotopes as a proxy for OM provenance occurred from 2,000-1,400 yr BP and from 1,100-750 yr BP (Knudson et al., 2011). These two periods broadly overlap with our interpreted strengthening of the winds from downcore $\delta^{13}$C and C/N of bulk sediment, and from benthic foraminifer stable isotopes (Fig. 9). Based on this comparison, we

interpret that the westerlies were strong over ~6° of latitude within the New Zealand sector of the southwest Pacific Ocean from 1,600-1,400 and 1,100-900 yr BP. Weakened wind strength in Fiordland from 1,400-1,100 yr BP while westerlies were still strong at the Auckland Islands (an anti-phased relationship) argues for a southward shift or contraction in SHWW, rather than a change in intensity at this time. Weakening of westerly circulation over the Auckland Islands between 1,000-900 yr BP (Turney et al., 2016 and current study) occurred a few centuries prior to weakening over Fiordland (Knudson et al., 2011), suggesting a northward

migration of the SHWW at this time. This northward migration coincided with the end of the Northern Hemisphere expression of the Medieval Climate Anomaly (MCA; 1,000-700 yr BP) and the beginning of the Little Ice Age (LIA; 500-100 yr BP; Masson-Delmotte et al., 2013), and is likely related to a shift to more negative phase of the Southern Annular Mode (SAM) (Moreno et al., 2014; Koffman et al., 2014; Abram et al., 2014; Villalba et al., 1997, 2012; Ummenhofer et al., 2009).

**5.4.2 Southern South America**

Select SHWW records from 50-53° S are compared with the current record (51° S) in order to assess zonal symmetry during the middle to late Holocene (Fig. 9). A $\delta^{18}$O record from fine-grained (<63μm) biogenic carbonate recovered in a sediment core from Lago Guanaco, Chilean Patagonia (51° S, 73° W) indicates increased wind-driven evaporation from ~1,600 yr BP (Fig. 9; Moy et al., 2009). An additional westerly wind construction from Lago Cipreses at the same latitude and longitude uses a *Nothofagus*/Poaceae Index (NPI) record to indicate vegetation type, which is directly related to the wind-driven precipitation

(Moreno et al, 2014). Dominance of tall *Nothofagus* forest indicating increased SHWW influence occurs from 1,700-1,400 yr BP and 1,000-800 yr BP, similar in timing to intensified winds in Fiordland (Knudson et al., 2011; Fig. 9). In addition, a pollen record from the Falkland Islands (52° S, 58° W) indicates intensified winds from 2,000 to 1,000 yr BP (Turney et al., 2015).

At 52° S (73° W), a carbonate $\delta^{18}$O record from a stalagmite recovered from the Marcelo Arévalo in Cave in southwest Patagonia

(MA-1; Schimpf et al., 2011) indicates increased wind-derived precipitation from 1,500-800 yr BP, and decreased precipitation from 800-500 yr BP (Fig. 9), consistent with our record from the Auckland Islands. Another study from a fjord system in the Strait of Magellan (53° S, 70° W), southern Patagonia, reconstructs Holocene changes in westerly wind-derived precipitation and associated changes in salinity and productivity using accumulation rates of biogenic silica, organic carbon, biogenic carbonate,

and siliciclastics (Aracena et al., 2015). Increased siliciclastic, organic carbon (both marine and terrestrial) and carbonate AR at ~2 ka likely indicates an intensification of freshwater flux from increased precipitation (hence SHWW intensification) at this time. Slightly elevated glacial clay content, increased terrestrial organic carbon AR and a significant decrease in carbonate accumulation rate from 3.2-2.2 ka are inferred to reflect another period of intensified westerlies.

Overall these records indicate a late Holocene intensification of the SHWWs across southern South America (at least south of ~50° S), which coincides with SHWW intensification at the subantarctic Auckland Islands, suggesting that the winds behaved in a zonally consistent manner across the Pacific Ocean. This in turn implies that climate variability in both locations was controlled by a common forcing mechanism, such as atmospheric teleconnections between the low and high latitudes of the Pacific Ocean (Haug et al., 2001; Mayweski et al., 2004; Villalba et al., 1997).

### 5.4.3 West Antarctica

Dust particles in the West Antarctic Ice Sheet (WAIS) Divide ice core (79° S, 112° W) record changes in mid-latitude SHWW circulation over the past 2,400 years (Koffman et al., 2014). An increase in percentage of coarse particles occurs at 1,850 and 1,650 yr BP and more recently from 900-550 yr BP (similar in timing to the MCA). These millennial-scale periods of intensified SHWWs are roughly coincident with late Holocene intensification at the subantarctic Auckland Islands. Comparisons of the WAIS Divide dust flux record to other Pacific paleoclimate records indicates that southward shifts of the westerlies are associated with changes in solar irradiance (Steinhilber et al., 2009), and increasing strength of the El Niño Southern Oscillation (ENSO; Moy et al., 2002; Yan et al., 2011), indicating that the tropical Pacific exerts a strong influence on high-latitude climate through atmospheric teleconnections.

Upper ocean temperatures along the western Antarctic Peninsula (WAP) are largely modulated by SHWW-driven upwelling of relatively warm (~2° C) modified Circumpolar Deep Water (CDW) onto the continental shelf (Martinson et al., 2008; Martinson and McKee, 2012), hence ocean temperature reconstructions from this area can be interpreted in terms of wind strength (Shevenell et al., 2011). With increased SHWW influence (increased strength/ southward position), upwelling of CDW increases which warms shelf waters along the WAP. An upper ocean (0-200 m) temperature record derived using the TetraEther indeX of 86 carbon atoms (TEX$_{86}$) from Palmer Deep (64° S, 64° W; Fig. 9) demonstrates warming around 1,600 yr BP, indicative of increased SHWW influence at this latitude, associated with initiation of atmospheric teleconnections between the WAP and tropical Pacific (Shevenell et al., 2011).

Low- to high-latitude atmospheric teleconnections modulated by ENSO may also influence wind variability in the southwest Pacific over the late Holocene. For example, SSTs and presumably $\delta^{13}C_{DIC}$ in surface water masses around New Zealand are also affected by zonal circulation patterns via anomalous Ekman transport and heat fluxes, associated with fronts of the wind-driven ACC (Ummenhofer and England, 2007). Modelling studies described above demonstrate that the modern southward shift of the SHWW belt is manifested in increased SST and decreased productivity in the SAZ, therefore this process could be partially controlling surface water $\delta^{18}O$ and $\delta^{13}C_{DIC}$ over the southwest New Zealand continental shelf. This connection suggests that there are wide-spread changes in physical oceanography related to SHWW variability over the late Holocene.

In summary, SHWW records from Patagonia (51-53° S), the WAIS Divide (79° S) and the WAP (64° S) argue for increased wind strength during the late Holocene (onset between 2,000 and 1,600 yr BP), and new and existing (McGlone et al., 2000; Turney et

al., 2016) evidence from the Auckland Islands demonstrates a broadly contemporaneous increase in wind strength, supporting the argument that the SHWW have behaved in a zonally consistent manner from at least 1,600-900 yr BP. However, before 2,000 yr

BP, antiphased relationships in wind strength at comparable latitudes including the current record, implies asymmetry in the SHWW during the middle Holocene (Fletcher and Moreno, 2012). However, it should be noted that interpretations of SHWW variability are complicated due to varying sensitivity of different proxies, and the contradictory nature of using multiple proxies with poorly defined relationships to the modern environment (see discussions by Kilian and Lamy, 2012 and Fletcher and Moreno, 2012).


Taken together, our record of middle to late Holocene sub-millennial-scale SHWW variability from the Auckland Islands and others from a range of latitudes across the Pacific Ocean indicate a southward shift during the MCA (1000-750 yr BP) and a northward shift at the start of the LIA (500-100 yr BP). This is important because zonal symmetry and latitudinal migrations of the SHWWs not only affect regional precipitation patterns and climate, but influence global carbon cycling, whereby a northward

(southward) position of the SHWW is generally associated with decreased (increased) atmospheric $CO_2$ on longer timescales (Anderson et al., 2009; Sigman et al., 2010; Skinner et al., 2010; Toggweiler et al., 2006). Under the most recent change to a predominately positive phase of the SAM, largely as a result anthropogenic greenhouse gas emissions and stratospheric ozone layer depletion (Shindell and Schmidt, 2004; Thompson and Solomon, 2002), the SHWWs are expected to remain in a southerly position with implications for ongoing atmospheric $CO_2$ rise and atmospheric warming.

**6 Conclusions**

 We reconstructed past changes in the strength of the SHWW using a sediment core recovered from a silled inlet at the Auckland Islands, located in the SAZ south of mainland New Zealand, within the modern SHWW core. Modern process studies reveal that increased summer precipitation and wind strength induces rapid and complete vertical mixing of fjord waters, reduced fjord salinity, and triggers return flow of saline shelf water rich in $OM_{mar}$ at depth.


Downcore changes in magnetic susceptibility, oxygen and carbon stable isotopes of benthic foraminifera, and bulk organic carbon isotopes and atomic C/N provide evidence for the Holocene paleoenvironmental evolution of Hanfield Inlet. Seismic data and a transition from lacustrine to marine sediments reveals the post-glacial evolution of Hanfield Inlet, where marine incursion into a paleolake occurred around 10,000- 9,000 yr BP, based on fjord entrance sill height and local sea level reconstructions (Clement et

al., 2016; Dlabola et al., 2015). Magnetic susceptibility, bulk organic $\delta^{13}C$ and atomic C/N indicate increased SHWW influence from 1,600-900 yr BP, consistent with previous palynological reconstructions of precipitation and wind strength at the Auckland Islands (McGlone et al., 2000; Turney et al., 2016). Bulk organic $\delta^{13}C$, and wt% TN and TOC indicate increased productivity from 900-500 yr BP, associated with increased fjord stratification due to decreased influence of SHWW. Comparison of the Auckland Islands record of SHWW with another from Fiordland, New Zealand (Knudson et al., 2011) indicates that the SHWW field shifted

southward from ~1,400-1,000 yr BP and northward towards southern New Zealand at ~900 yr BP. From ~900-750 yr BP, roughly coincident with the end of the MCA (1000-700 yr BP; Masson-Delmotte et al., 2013), the winds contracted further north and this was likely associated with dominance of the negative phase of the SAM during this time (Abram et al., 2014; Moreno et al., 2014; Villalba et al., 1997, 2012). Comparison with records from 51-53°S in southern South America (Aracena et al., 2015; Moreno et al., 2014; Moy et al., 2009; Schimpf et al., 2011; Turney et al., 2016), and West Antarctica (Koffman et al., 2014; Shevenell et al.,

2011) indentifies a late Holocene intensification of the SHWW, indicating that the winds may have been zonally symmetric across the Pacific Ocean from at least 1,600-900 yr BP, and asymmetric after this.

Sub-millennial scale variations in the stable oxygen and carbon isotopes of benthic foraminifera from Hanfield Inlet are controlled indirectly through the influence of SHWW on shelf and fjord hydrography. The strength, symmetry and latitudinal position of the
SHWW (modulated by both the SAM and ENSO) exert a dominant control on upwelling of nutrient- and carbon- rich water masses along fronts of the ACC, which are exported northwards to the New Zealand continental shelf. More negative foraminifer $\delta^{13}C$ and $\delta^{18}O$ are related to increased wind-derived fjord precipitation (decreased salinity and decreased $\delta^{13}C_{DIC}$ due to increased respiration of $OM_{terr}$), and input of warm shelf waters with more negative $\delta^{13}C_{DIC}$ as a result of enhanced wind-driven upwelling and northward export of SAMW south of New Zealand (Lovenduski and Gruber, 2005; Lovenduski et al., 2008; Wetzel, 2005). Intensified
SHWW during the late Holocene coincides with increased teleconnections between the tropical Pacific and high latitudes, and this study provides the first evidence for such teleconnections being present in the southwest Pacific Ocean.

**7 Author Contribution**

I. Browne, C. Moy, L. Curtin, A. Gorman, and G. Wilson carried out field work. C. Moy, C. Riesselman and I. Browne planned the experiments and contributed to data interpretation. I. Browne and L. Curtin carried out stable isotope analysis of water,
particulates, and sediment. I. Browne and H. Neil carried out foraminifer stable isotope analysis. I. Browne prepared the manuscript with contributions from all co-authors.

**8 Competing Interests**

The authors declare that they have no conflict of interest.

**9 Acknowledgements**

The authors gratefully acknowledge support from a Royal Society of New Zealand Marsden Fast Start (#UOO1118) and New Zealand Antarctic Research Institute (NZARI 2014-7) funding to C. Moy. Seismic data were processed using an academic licence for GLOBE Claritas by B. Ross and A. Gorman. NCEP Reanalysis data provided by the NOAA/OAR/ESRL PSD, Boulder, Colorado, USA, from their website at http://www.esrl.noaa.gov/psd/. We would like to thank B. Hayward for help with identification of benthic foraminifera, T. Max at NIWA for analysing foraminifer stable isotopes, and R. Van Hale, K. Currie and
C. Aebig at the University of Otago for analysing the isotopic composition of water, particulates and sediment. We thank C. Aracena and an anonymous reviewer for helpful comments on an earlier version of the manuscript. We also thank P. Moreno for providing pollen data from Lago Cipreses, and R. Kilian for providing MA-1 $\delta^{18}O$ data. Special thanks to the crew and scientific party on Auckland Islands cruises 14PL001 and 15PL001, the Young Blake Expedition members, and B. Dagg, whose coring expertise was fundamental to the achievement of this work.

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

**Table 1. Median probability ages and associated error for eight AMS radiocarbon-dated terrestrial organic macrofossil fragments from 36P4, Hanfield Inlet. Radiocarbon years were converted to cal yr BP using SHCal13 (Hogg et al., 2013) in Calib 7.0.4 (Stuvier and Reimer, 2003).**

| # | CAMS ID | Sample ID | Depth (cm) | $^{14}$C years | Median Probability Age (cal yr BP) | Lower 2θ | Upper 2θ |
|---|---------|-----------|------------|----------------|-------------------------------------|----------|----------|
| 1 | 167798 | OU_2014_57 | 12.5 | 470 ± 30 | 500 | 340 | 526 |
| 2 | 167767 | OU_2014_13 | 58 | 970 ± 80 | 830 | 685 | 956 |
| 3 | 168241 | OU_2014_74 | 126.5 | 2180 ± 70 | 2136 | 2017 | 2299 |
| 4 | 168242 | OU_2014_75 | 225 | 3560 ± 210 | 3820 | 3512 | 4089 |
| 5 | 167768 | OU_2014_14 | 284 | 4000 ± 40 | 4420 | 4255 | 4525 |
| 6 | 167769 | OU_2014_15 | 366 | 4190 ± 30 | 4690 | 4539 | 4827 |
| 7 | 168243 | OU_2014_76 | 451 | 5360 ± 140 | 6095 | 5942 | 6272 |
| 8 | 167799 | OU_2014_62 | 542.5 | 5315 ± 30 | 6060 | 5938 | 6180 |

**Table 2. Locations of surface sediment samples and accompanying bottom water oxygen ($\delta^{18}O_w$) and carbon ($\delta^{13}C_{DIC}$) stable isotope measurements. Expected oxygen isotopic composition of calcite precipitated in equilibrium with $\delta^{18}O_w$ ($\delta^{18}O_{foram}$) for a given temperature was calculated using the equation from Marchitto et al., 2014 ($\delta^{18}O_{foram}$-$\delta^{18}O_w$(VSMOW)+0.27)= -0.225T +3.50).**

| Cruise | CTD | Site/sample | Fjord | Height of measurement above substrate (m) | $\delta^{18}O_w$ (‰) (VSMOW) | $\delta^{13}C_{DIC}$ (‰) (VPDB) | Temperature (°C) | $\delta^{18}O_c$ (‰) (VPDB) |
|--------|-----|-------------|-------|---------------------|------|------|------|------|
| 14PL001 | CTD_001 | 35G1 | Norman | 1 | -0.26 | 0.64 | 10.55 | 0.60 |
| 14PL001 | CTD_002 | 36G1, 36B2, 39G1 | Hanfield | 4 | -0.1 | 1.21 | 10.55 | 0.76 |
| 14PL001 | CTD_004 | 36G1, 36B2, 39G1 | Hanfield | 8.5 | -0.24 | 0.03 | 10.5 | 0.63 |
| 15PL001 | CTD_006 | - | Norman | | 0.31 | - | 10.4 | 1.31 |
| 15PL001 | CTD_007 | - | Hanfield | | 0.29 | - | 9.9 | 1.28 |

**Table 3. Average benthic foraminifer oxygen and carbon isotopic compositions ($\delta^{18}O_{foram}$ and $\delta^{13}C_{foram}$) from grab and box-core**
**sediments collected from Hanfield (sites 36 and 39) and Norman Inlet (site 35) in 2014.**

| Species | Number of duplicate measurements | Average $\delta^{18}O_{foram}$ (‰) VPDB | Standard deviation for $\delta^{18}O_{foram}$ (‰) VDPB | Av $\delta^{13}C_{foram}$ (‰) (VPDB) | Standard deviation for $\delta^{13}C_{foram}$ (‰) VDPB |
|---------|-----------|---------|---------|---------|---------|
| *B. marginata* f. *marginata* | 5 | 1.64 | 0.10 | -0.34 | 0.31 |
| *Cibicides* spp. | 3 | 1.25 | 0.07 | 1.67 | 0.12 |
| *N. flemingi* | 5 | 2.00 | 0.21 | -0.99 | 0.49 |
| *Q. seminula* | 6 | 1.60 | 0.22 | -0.11 | 0.31 |
| *T. angulosa* | 6 | 1.66 | 0.16 | 0.49 | 0.25 |

**Table 3. Median Probability Ages and associated error for eight AMS radiocarbon-dated terrestrial organic macrofossil fragments from 36P4, Hanfield Inlet. Radiocarbon years were converted to cal yr BP using SHCal13 (Hogg et al., 2013) in Calib 7.0.4 (Stuvier and Reimer, 2003).**

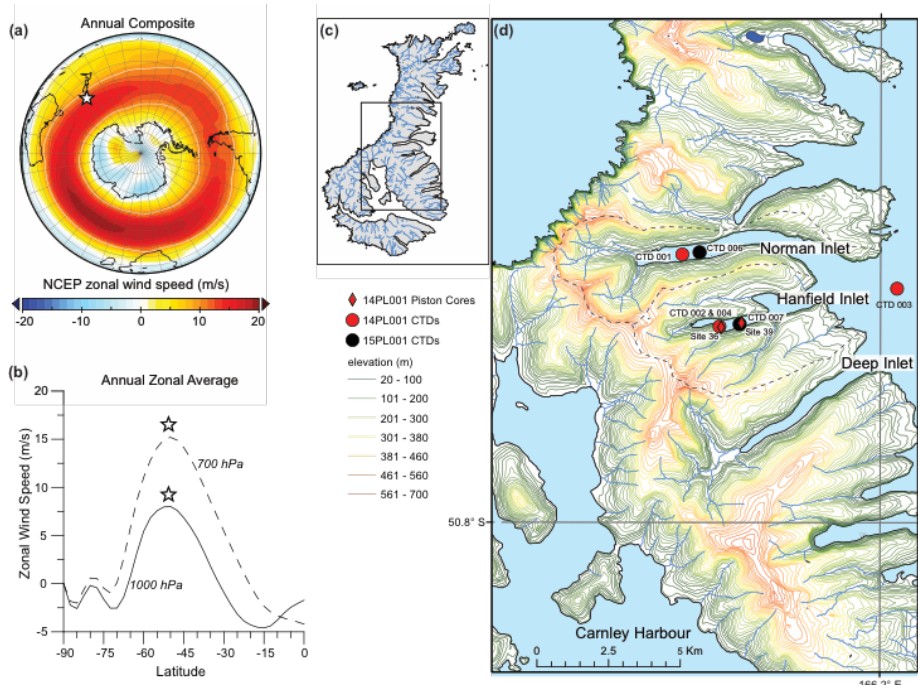

**Figure 1. (a) and (b) Annual average of 700 hPa zonal wind speed. Data sourced from the NCEP/NCAR reanalysis (Kalnay et al., 1996). The star denotes the position of New Zealand's subantarctic Auckland Islands (51° S, 166° E). (c) Map of Auckland Islands with study area highlighted. d) Piston coring (diamonds) and CTD (circles) sites. Core 36P4 was collected at Site 36 and Core 39P4 was collected at Site 39. Data were collected in February of 2014 (cruise 14PL001, red symbols) and 2015 (cruise 15PL001, black symbols). Dashed black lines outline the catchment areas for Norman and Hanfield Inlets.**


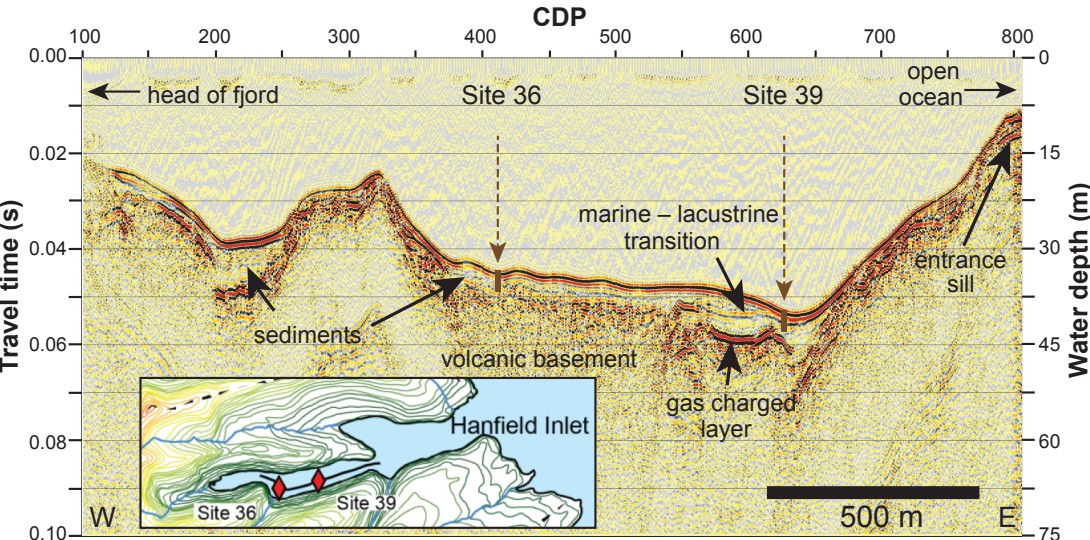

**Figure 2. High-resolution boomer migrated seismic line 14PL001-25 through Hanfield Inlet, with location as shown on inset map. The fjord contains two sub-basins containing sediments of variable thickness, and a shallow entrance sill with a minimum depth of ~8 m. A low-amplitude continuous reflection in the eastern basin probably corresponds to the marine to lacustrine transition sampled in core 39P4 (site 39). Also in the eastern basin, a deeper and stronger reflection, with a polarity opposite that of the seafloor, probably corresponds to a gas charged layer. The core used for paleoclimate reconstruction is 36P4 (site 36). Beneath the basins, the basement volcanic rocks appear mostly to be non-reflective.**


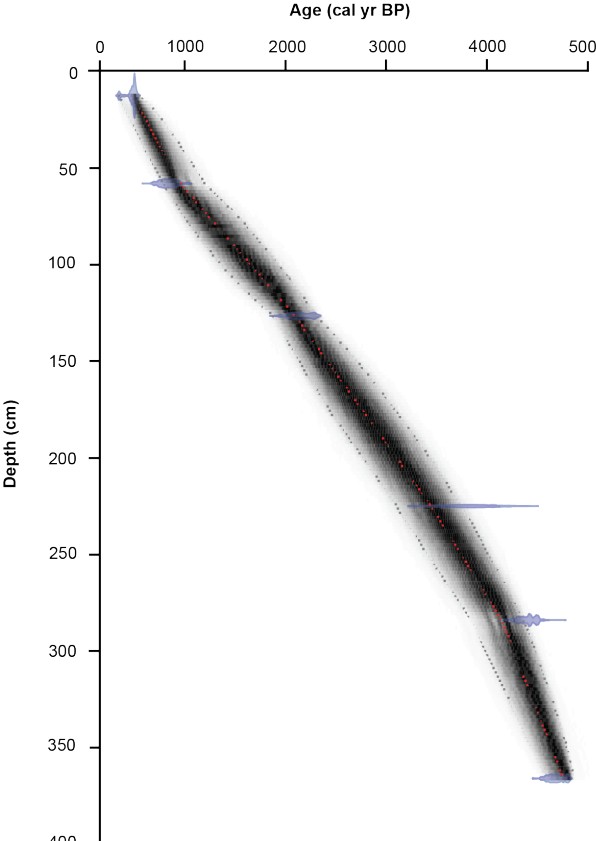

**Figure 3. A posterior age-depth model for piston core 36P4 (grey field) calculated using Bayesian interpolation of six [14]C dates (from terrestrial organic macrofossil fragments) from core 36P4 from Hanfield Inlet. Age-depth model was constructed using Bacon software (Blauuw and Christen, 2011). See text for model input parameters. Radiocarbon ages were calibrated to cal yr BP using SHCal13 (Hogg et al., 2013). Red dots indicate the weighted mean calibration age and the grey dots represent the 95% confidence interval.**

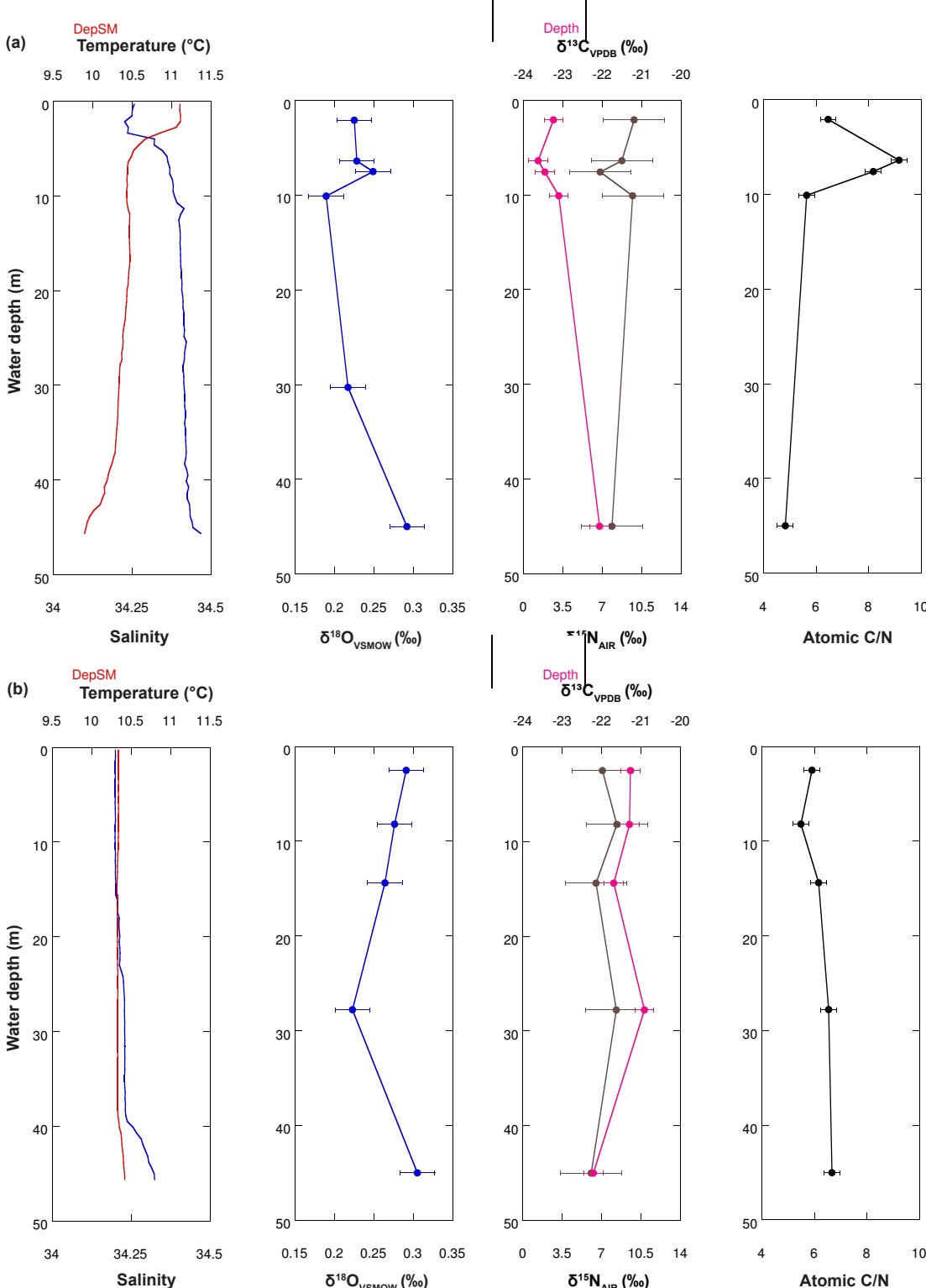

Figure 4. Profiles of modern fjord water temperature, salinity, $\delta^{18}O$ and $\delta^{13}C$, $\delta^{15}N$ & atomic C/N of acidified particulate organic matter (OM) sampled in February 2015, representing end-member synoptic weather conditions. a) CTD_007 (Hanfield Inlet) was sampled when a high-pressure system had been situated over the Auckland Islands for >24 hours and is representative of profiles collected under fair weather conditions (low wind velocities, no precipitation). Under these conditions, the water column exhibits weak stratification. Temperature is slightly elevated and salinity is slightly depressed in the upper 4 m of the water column, and $\delta^{18}O$ is more positive in the upper ~10 m. Particulate OM $\delta^{15}N$ shows an overall increase with depth, while $\delta^{13}C$ and molecular C/N show a decrease. b) CTD_006 (Norman Inlet) was sampled when a low-pressure system was in the vicinity of the Auckland Islands, producing high W/SW wind velocities and precipitation. Under these conditions, the water column is completely mixed. Temperature and salinity are largely invariant with depth and increase only slightly below 40m, while particulate OM $\delta^{13}C$, $\delta^{15}N$ & atomic C/N are invariant within error. Error bars are the average standard deviation of duplicates for each CTD cast ($\delta^{18}O$) and for all filtered particulates ($\delta^{13}C$, $\delta^{15}N$ & C/N).

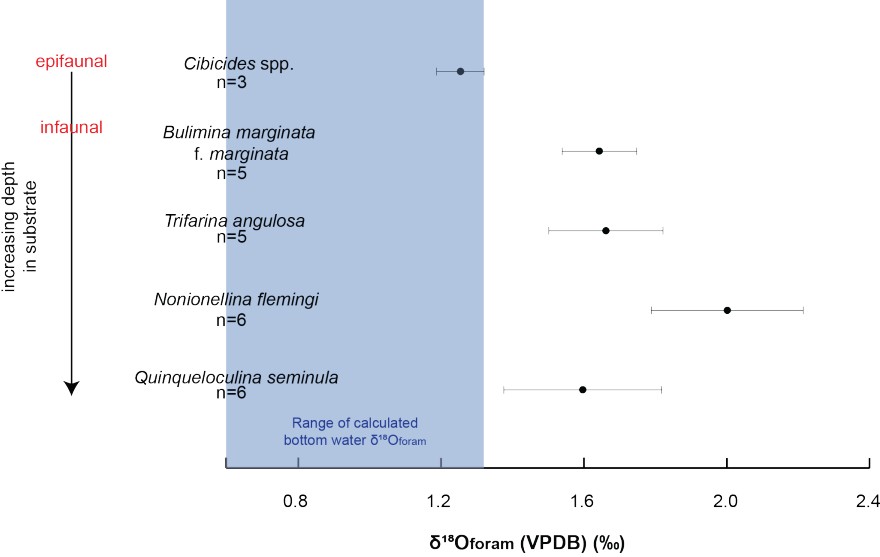

**Figure 5. Deviation of benthic foraminiferal oxygen isotopic composition ($\delta^{18}O_{foram}$) from calculated equilibrium according to Marchitto et al., 2014 ($\delta^{18}O_{foram}$-$\delta^{18}O_w$(VSMOW)+0.27)= -0.225T +3.50) for five species from Hanfield and Norman Inlets. This equation, derived for *Cibicides* spp. is indistinguishable from the temperature equation from Kim and O'Neil (1997) for inorganic calcite, so is considered valid for all species. Range of expected $\delta^{18}O_{foram}$ encompasses the entire range of measured summer fjord bottom water temperatures and $\delta^{18}O_w$ from Hanfield and Norman Inlets (Tables 2 and 3).**

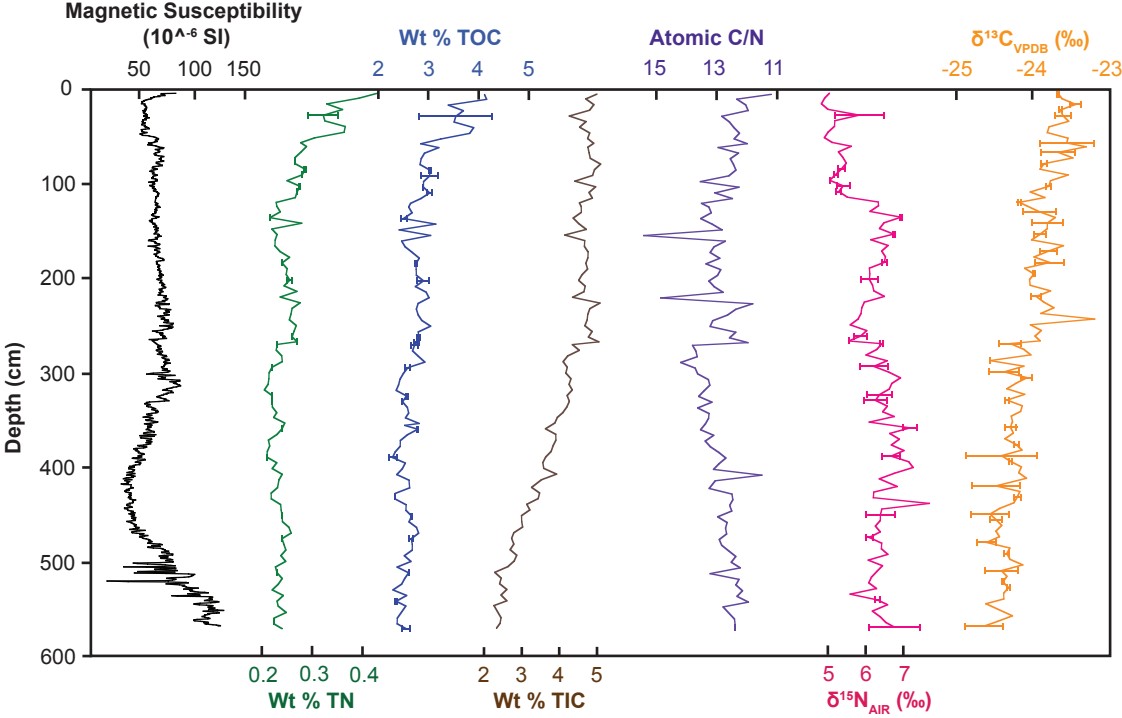

**Figure 6. Downcore sedimentary parameters plotted against depth for core 36P4 collected in Hanfield Inlet: magnetic susceptibility, wt % nitrogen (TN), wt % organic carbon (TOC), wt % carbonate (TIC), $\delta^{13}C$, $\delta^{15}N$ and atomic C/N of bulk sediment (note reversed scale). Error bars are 2 σ for duplicate measurements.**

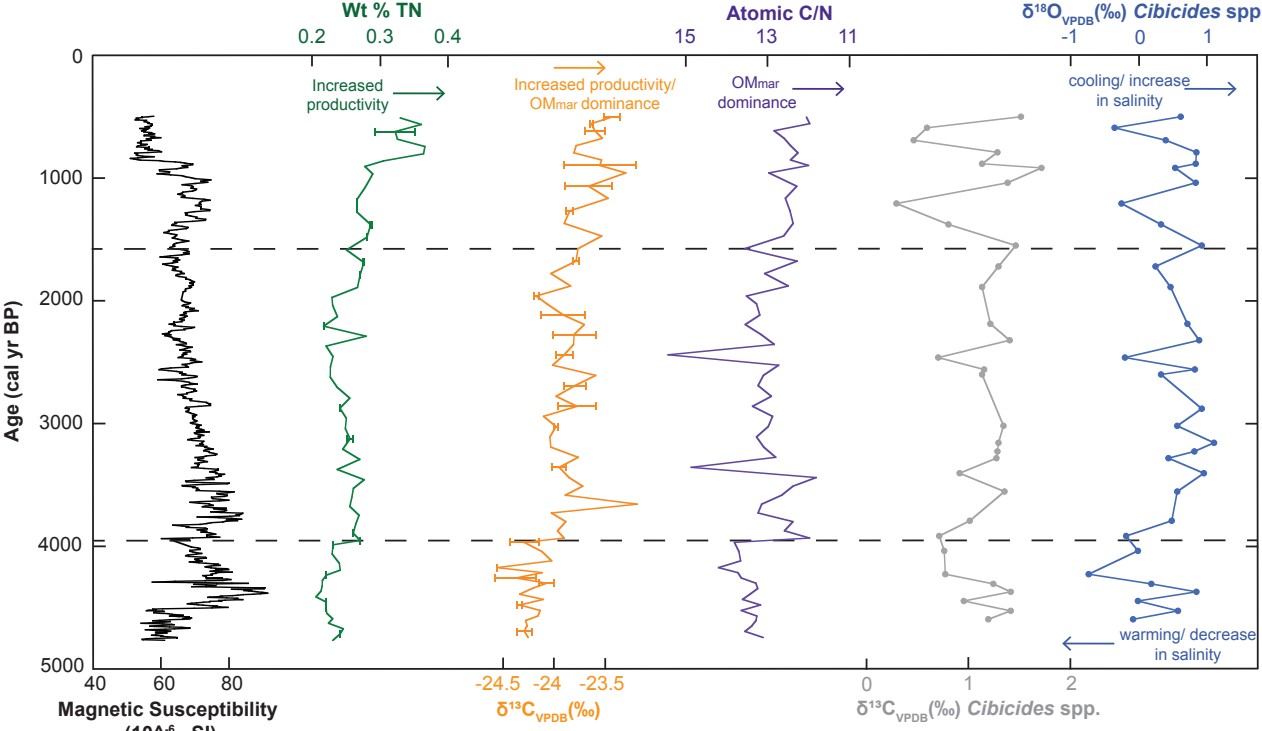


**Figure 7. Downcore parameters against age, 36P4, Hanfield Inlet: magnetic susceptibility (MS), wt % nitrogen (TN), $\delta^{13}$C and atomic C/N of bulk sediment, and $\delta^{13}$C & $\delta^{18}$O of benthic foraminifer *Cibicides* spp. Error bars on the bulk organic C and N data are 2 σ for replicate measurements (see text for stable foraminifer isotope error). The dashed horizontal lines at ~4,000 and 1,600 yr BP break the core into three sections based on prominent shifts across multiple proxies. From 4,000-1,600 yr BP, fjord conditions are relatively stable,**

**as indicated by relatively unchanging MS, wt % TN, $\delta^{13}$C and atomic C/N. This interval is characterised by more negative $\delta^{13}$C and higher C/N, indicating a higher contribution from OM$_{terr}$, increased stratification and decreased SHWW influence at this time. From 1,600-900 yr BP, increasing wt %TN, more positive $\delta^{13}$C and more negative C/N indicates a shift to OM$_{mar}$ dominance, signifying a strengthening of the SHWW at this time. Paired fluctuations in benthic foraminifer $\delta^{13}$C and $\delta^{18}$O demonstrate increased variability during this period, associated with westerly-wind driven changes in shelf hydrography. From 900-500 yr BP, a decrease in MS**

**accompanied by an increase in wt % TN indicates an increase in productivity (also reflected by more positive $\delta^{13}$C), rather than strengthened winds at this time.**

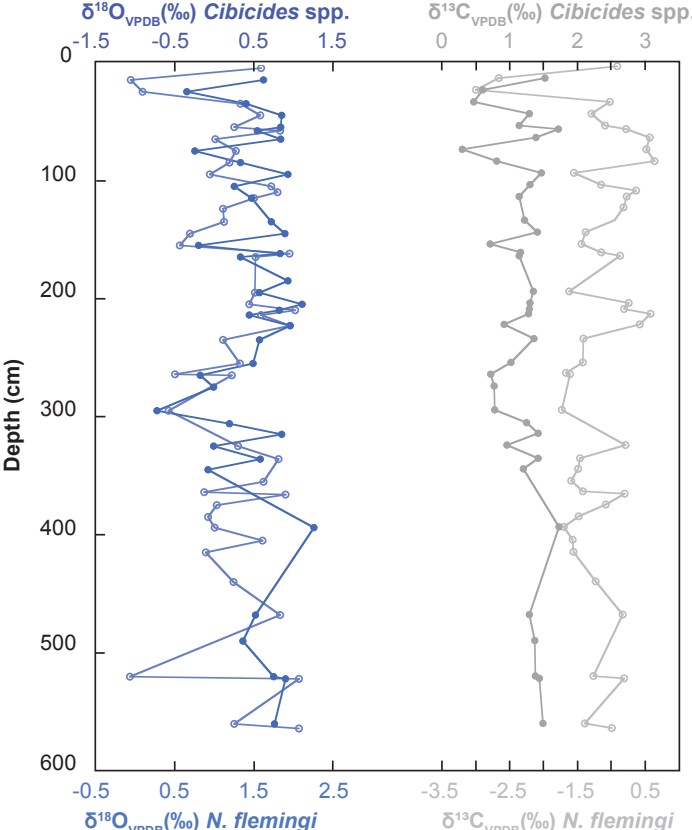

**Figure 8. Downcore variations in $\delta^{18}O_{foram}$ for oxygen (left) and carbon (right) isotopes, 36P4, Hanfield Inlet. Solid circles represent *Cibicides* spp., and open circles represent *N. flemingi*. Both species show similar trends and similar magnitude of $\delta^{18}O_{foram}$ variation in downcore, although *N. flemingi* exhibits more positive values relative to *Cibicides* spp. (due to species-specific vital offsets; see Fig. 5). Both the trends and absolute magnitude of isotopic shifts for $\delta^{13}C_{foram}$ differ between the two species. *N. flemingi* exhibits more negative $\delta^{13}C_{foram}$ with a higher range of values, whereas *Cibicides* spp. exhibits more positive $\delta^{13}C_{foram}$ with a lower range. Note: we focus our paleoclimate interpretations on the stratigraphic interval above 420cm.**


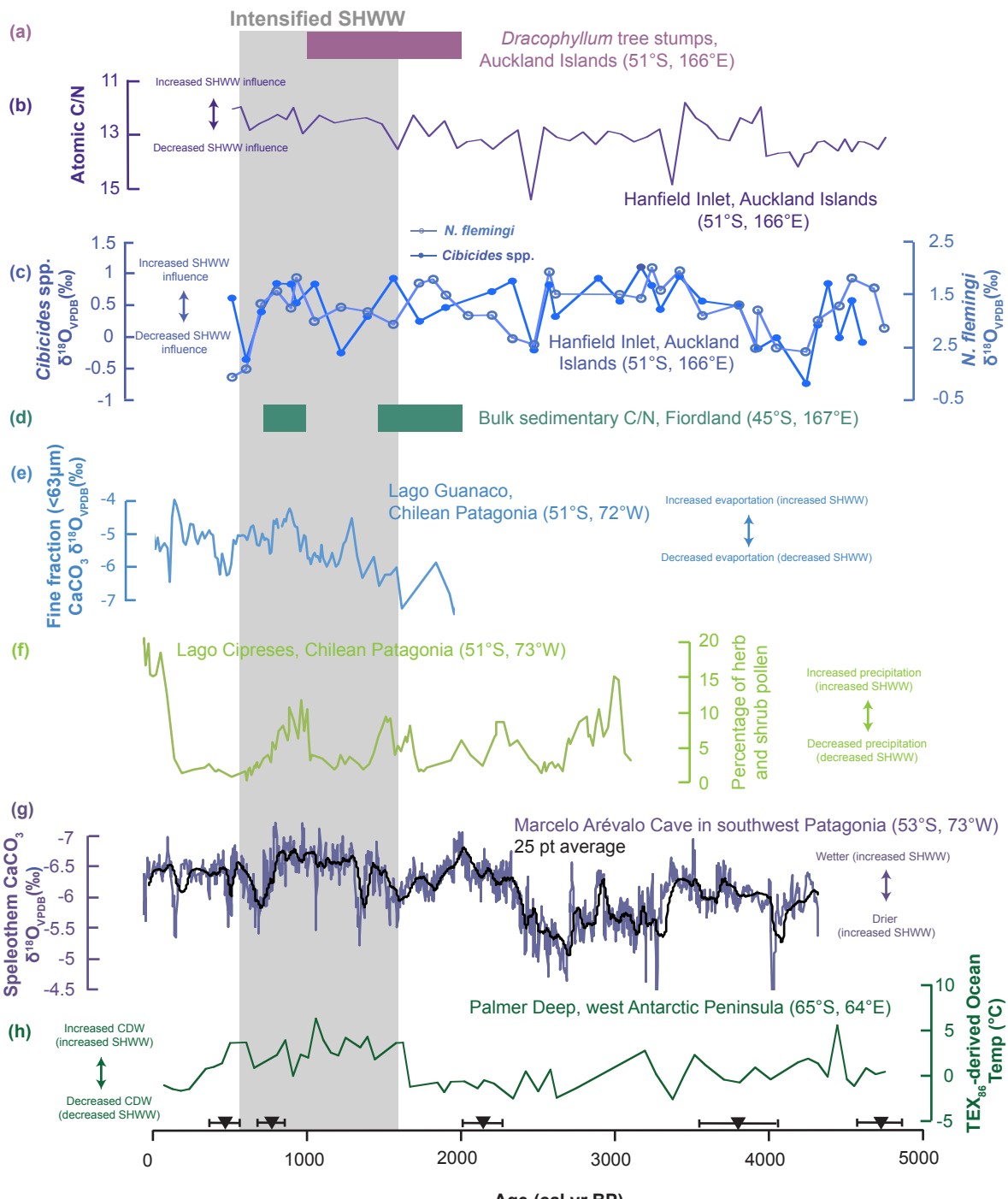

Figure 9. Compilation of Southern Hemisphere paleoclimate records illustrating SHWW variability during the middle and late Holocene. Horizontal coloured bars indicate periods of intensified westerly winds and the vertical grey bar represents late Holocene intensification of the winds in several areas. Black triangles along the x-axis represent the median probability ages with calibration error for 36P4. (a) Tree-line reconstruction using *Dracophyllum* tree stumps from the Auckland Islands (Turney et al., 2016). (b) Bulk sedimentary C/N from Hanfield Inlet, Auckland Islands. (c) Benthic foraminifer $\delta^{18}O$ (*Cibicides* spp. and *N. flemingi*) from Hanfield Inlet, Auckland Islands. (d) Bulk sedimentary atomic C/N from Fiordland, New Zealand (Knudson et al., 2011). (e) Fine fraction (<63 μm) biogenic carbonate $\delta^{18}O$ from Lago Guanaco, Chilean Patagonia (Moy et al., 2009). (f) Pollen record from Lago Cipreses, Chilean Patagonia (Moreno et al., 2014). (g) Stalagmite carbonate $\delta^{18}O$ from Marcelo Arévalo Cave in southwest Patagonia (Schimpf et al., 2011) (h) TEX$_{86}$-derived upper ocean temperatures (0-200 m) from the Palmer Deep, western Antarctic Peninsula (Shevenell et al., 2011).