# Peer review of "Late Holocene intensification of the westerly winds at the subantarctic Auckland Islands (51° S), New Zealand"

_Climate of the Past, 2017_

## Referee Comment (RC1) · Anonymous Referee #1 · 1 Jun 2017

See supplementary file

Please also note the supplement to this comment:
http://www.clim-past-discuss.net/cp-2017-52/cp-2017-52-RC1-supplement.pdf

---

## Referee Comment (RC2) · C. Aracena (Referee) · 7 Jun 2017

Dear Imogen,

Your manuscript "Late Holocene intensification of the westerly winds at the subantarctic Auckland Islands (51°'S), New Zealand" is a very interesting contribution for the knowledge of past climate, and ocean reconstructions at the Southern Hemisphere. In general, the quality of the information described in the manuscript is good, as well as the figures and tables. With humility and to enhance the writing of your scientific contribution my main suggestions are: 1) to include in the study area climatological modern data description related to precipitation and runoff discharge if exists, and 2)

[Figure]

re-organize the discussion, and results chapter. The discussion is too long and not focused on the topic described in the title of this manuscript. I believe that the chapters 5.4.1, 5.4.2, and 5.4.3 could be nicely summarized in a discussion table.

Also there are minor changes to be done in order to highlight the results of this interesting investigation.

1. Between lines 15 and 25: You mention "Drainage basin response" ,"Hydrographic response", and "vegetation response". All in relation to SHWW variability. I believe that is necessary to clarify such "responses" because is too vague. This would make easier for the reader to have a better comprehension of what you want to say when you mention such "responses".

2. There is an inconsistency with the term between the meaning of C/N lines 18, and 83. ..."monitor influxes of lithogenous, and terrestrial vs marine organic matter"... Lithogenous has nothing to do with organic matter provenance I believe.

3. Between lines 87, and 94: I suggest addressing these questions to the discussion chapter. In the introduction I would only leave open the question that can be fully addressed by the results presented in this manuscript.

4. First paragraph between lines 117 to 121: I suggest to move this historical information to the introduction chapter.

5. Between lines 121 to 125: I suggest to move this paragraph to continue in line 103 as.... At the present, shelf waters surrounding the Auckland....

6. Paragraph starting in line 123 to line 127: I suggest to move to the introduction chapter, considered as an author's hypothesis that also justifies this study.

7. In the Methods chapter I suggest to move lines 137 and 140 (about seismic profiles results) to the result chapter of the manuscript.

8. In the discussion chapter I suggest to move lines from 284 to 294 somewhere to the

introduction chapter or include into the study area as a regional setting.

9. Between lines 299 to 300 you mention "we speculate". Also use this starting in sentence in line 318. I suggest for the discussion chapter starting sentences that enhance the value of the data presented in this investigation.

10. Error in line 471 because the cited reference does not describe inorganic carbon, it describes biogenic carbonate.

11. Figure 1. There is a universe of stars in the figure. In general, I suggest resizing (to observe detailed view of piston corer as you wrote in the captions), and labelling again the pannels because there are two figures more, not only one. Is necessary to indicate site 36P4 in bigger pannel ( c). Would be interesting to include modern climatological data in the area (for example, precipitation curves and/or runoff discharge) since the interpretation of sediments is based on organic matter provenance proxies. I suggest to use another forms and colors to make differences on what is written in the text. See other error on figure: rhombus overlapping red star in the bigger ( c) pannel.

12. Figure 2. Is necessary to indicate site on figure sites 36P4 and 39P4.

13. Figure 7. In relation to interpretation on organic matter provenance I suggest to look at the paper of Perdue and Koprivnjak, 2007 where they use molar N to C ratios to define end members and discriminate between terrestrial and marine organic carbon.

I hope all these recommendations would improve the communication of your manuscript.

My best regards and success,

Claudia M. Aracena P.

---

## Author Comment (AC1) · 21 Jun 2017

We thank Claudia Aracena and an anonymous reviewer for their valuable comments and suggestions for improvement of the manuscript. Please find our complete responses to both reviewer comments attached as a pdf supplement.

Please also note the supplement to this comment:
http://www.clim-past-discuss.net/cp-2017-52/cp-2017-52-AC1-supplement.pdf
* * *

---

## Author Response (AR1)

[revised manuscript text omitted]

1190   **Re:** cp-2017-52 RC1

**Title:** Late Holocene intensification of the westerly winds at the subantarctic Auckland Islands (51ºS), New Zealand

**Authors:** Imogen Browne, Christopher Moy, Christina Riesselman, Helen Neil, Lorelei Curtin, Andrew Gorman, and Gary Wilson

Revision notes in response to reviewer comments are provided in blue text.

1195   **Anonymous Reviewer #1:** This paper presents the first fjord-based climate reconstruction from sub-Antarctic Auckland Islands. The data make an important contribution to the SHAPE initiative, providing a ca. 4000 year reconstruction from a climatically important yet data sparse region. The paper is well written and should be published following some corrections. The strengths of this paper lie in the fact the authors have undertaken a detailed modern, process based study to understand the fjord's circulation and how this may relate to the Southern Hemisphere westerly winds and

1200 precipitation at their site. They use this as a basis for interpreting the palaeo-record. This is followed by comparisons to other studies, which as currently presented are not as clear as they could be.

Please note: I do not have enough experience with stable isotope analysis to comment on this aspect of the paper.

1205   **Specific comments**

Core sub-division (lines 269-274): It appears the core was divided into four main periods and that this was done visually rather than statistically, for example by hierarchical cluster analysis. If possible, it would be better to use a more objective approach to divide the record into different periods, especially as there are times of little variation or where

1210 some proxies appear to be changing more than others, and it is somewhat subjective where to separate different parts. For example, one could also just divide the core into three zones (i.e. combine zones 2 and 3). We concur with the reviewer that there is scope to simplify our zonation scheme. Figure 7 has been adjusted to include only three zones, using high amplitude variations (>1‰) in the foraminifer $\delta^{13}C$ and $\delta^{18}O$ profiles to identify units within the chronostratigraphically-constrained portion of the record. We choose this approach because the relationship of these

1215 carbonate proxies to modern fjord processes is well-constrained. The revised zones are from ~4,800-4,000 yr BP (a period of high variability), from 4,000-1,600 yr BP (a period of low variability) and 1,600-500 yr BP (a period of high variability - top two zones combined). When we use this approach, we observe that other proxies exhibit shifts along these boundaries. The text in section 4.3 has been adjusted accordingly.

1220 Line 364 onwards: The authors refer to figures using depth and age-scales (Figures 6 and 7). To make it easier to follow please include the depths relating to the different zones identified (i.e. depths associated with ca. 4000-1600, 1600-900, 900-500 yr BP). Approximate depths (from Fig. 6) as they relate to time-zones (Fig. 7) have been added to the text in section 5.3 (260-115cm; 115-40cm; 40-0cm). The authors state there is little variation in the proxies from ca. 4000-1600 yr BP, however there are two distinct peaks (albeit based on one

1225 sample each) in C/N during this time. What are the causes of these? These peaks result from slight increases in wt %TOC, which are accompanied by equally slight decreases in wt % TN. Because this value is a ratio, the amplitude for these single-sample peaks appears more prominent than the individual proxy curves, which is why we do not give them a great deal of weight. These peaks may be due to a slightly greater contribution of terrestrial organic matter, which has elevated C/N values. Increasing sample resolution would be necessary to determine whether these

1230 excursions are meaningful.

Selection of records used for comparison: While there are changes in the proxies (Figures 6 and 7), some are quite subtle. To aid comparisons to other studies it would help if periods of interpreted stronger or weaker winds from this study were marked on Figure 9. We clarified text in the Figure 9 caption so that it is clear the vertical grey bar on that

1235 figure illustrates strengthened westerly winds from ~1,600 yr BP as interpreted from the majority of proxies included in the figure. Also include an arrow showing the interpretation in 9c. An interpretation arrow has been included (Fig. 9c) where up indicates increased shelf water-mass presence in the Auckland Islands fjords, associated with increased westerly wind influence in the southeast Pacific Ocean, and down represents decreased westerly influence. Clearer acknowledgement of the errors in the age-depth model and implications when comparing to other studies should be

1240 considered. There are six 14C dates for nearly 400 cm of core, which means relatively large uncertainties (Figure 3). An extra sentence has been added in the discussion in section 5.4.1 to highlight chronological uncertainty: "Inconsistencies in the records may be due to varying sensitivity of different proxies to westerly wind strength and/ or

chronological uncertainty". We have also added triangles along the x-axis of Figure 9 showing our calibrated ages and uncertainty.

1. Southern South America

The authors compare their record to two studies (Strait of Magellan, 53°S, Lago Guanaco, 51°S) because they are at comparable latitudes to the Auckland Islands (line 478). There are other records from southern South America for these latitudes that would be worth considering. Including, but not limited to:

(a) Lamy et al. (2010) Holocene changes in the position and intensity of the southern westerly wind belt. Nature Geoscience 3: 695–699 (53°S)

(b) Schimpf et al. (2011) The significance of chemical, isotopic, and detrital components in three coeval stalagmites from the superhumid southernmost Andes (53°S) as high resolution palaeo-climate proxies. Quaternary Science Reviews 30: 443–459 (53°S)

(c) Moreno et al. (2014) Southern Annular Mode-like changes in southwestern Patagonia at centennial timescales over the last three millennia. Nature Communications 5:4375 (Lago Cipreses 51°S)

(d) Turney et al. (2016) A 250-year periodicity in Southern Hemisphere westerly winds over the last 2600 years. Climate of the Past 12:189–200 (Falkland Islands 52°S). We are very grateful to the reviewer for bringing these records to our attention. We have followed their recommendation by incorporating the cave stalagmite $\delta^{18}O$ data from Schimpf et al., 2011 and the pollen record from Moreno et al., 2014 into Fig. 9 and the discussion. The study by Turney et al., (2016) is also incorporated into the discussion section.

While the authors say they compare their record to select records from a range of latitudes (line 468), the Strait of Magellan and Lago Guanaco are the only two referred to. I suggest looking at what studies are available north of 51°S and south of 53°S, to help support the authors' interpretations. Including, but not limited to:

(a) Lamy et al. (2001) Holocene rainfall variability in southern Chile: a marine record of latitudinal shifts of the Southern Westerlies. Earth and Planetary Science Letters 185: 369–382 (41°S)

(b) Borromei et al. (2009) Multiproxy record of Holocene paleoenvironmental change, Tierra del Fuego, Argentina. Palaeogeography, Palaeoclimatology, Palaeoecology 286: 1–16 (Las Cotorras 54°S)

(c) Mauquoy et al. (2004) Late Holocene climatic changes in Tierra del Fuego based on multiproxy analyses of peat deposits. Quaternary Research 61: 148– 158 (54°S). Again, we thank the reviewer for the suggested references. Because one of our objectives is evaluating symmetry in the SHWWs by comparing records from similar latitudes across the Pacific, we elect to limit our discussion to records from 50-53°S. We have modified line 545 to reflect this zonal focus.

2. Western Antarctic Peninsula

The authors refer to Shevenell et al. (2011) and their ocean temperature reconstruction. There are other Antarctic records that may be of use, such as Koffman et al. (2014, Centennial-scale variability of the Southern Hemisphere westerly wind belt in the eastern Pacific over the past two millennia. Climate of the Past 10: 1125–1144), who link WAIS Divide ice core dust to the westerlies. I suggest broadening this section to other regions of Antarctica to include such studies. As above we have followed the reviewers recommendation and have incorporated the study by Koffman et al., (2014) into section 5.4.3.

3. Assessment of SHWW symmetry during the LIA

It is not clear how the authors are defining the Little Ice Age. The record ends ca. 500 yr BP, however, it is after this time that most evidence for cooler conditions in the Australasian region exists (e.g. PAGES 2k consortium (2013) Continental-scale temperature variability during the past two millennia. Nature Geoscience 6: 339–346). I acknowledge no records from the sub-Antarctic are included in this work. Given the timing of the end of the record, I suggest removing this section and integrating any relevant studies mentioned into sections 5.4.1-5.4.3. We use the Northern Hemisphere timing of the LIA, beginning at 600 cal yr BP (after Osborn et al., 2006) to provide context for our Southern Hemisphere records, but Referee 1 is correct that our record ends shortly after the beginning of the LIA. Section 5.5 (Assessment of SHWW symmetry during the LIA) has been removed and relevant text has been moved to the end of section 5.4.1 and clarified.

Technical corrections: All corrections made.
Line 29: Remove SAM and ENSO – not needed in the Abstract.
Line 161: Remove 'and'.
Lines 161-163: Break up the sentence. For example – Seawater samples collected in 2014 were analysed for oxygen isotopes using a Picarro 2120 wave-length-scanned cavity ring-down spectrometer (WS-CRDS), in the Isotrace Laboratory at the University of Otago. The average standard deviation for 11 duplicate measurements (fjord, lake, and stream samples) was 0.027 ‰ for δ18O.

Line 242: Remove 'sediment samples'.
Line 244: ratios.
1305 Line 268: Remove 'for core 36P4'.
Line 364: Include MS in the list of generally unchanging proxies.
Line 414: Include '(AR)' after 'accumulation rates'.
Line 904: Change 'negative' to 'lower'.
Figures 8 and S3: Enlarge open circles or change the shape as it is hard to see them.
1310 Figures 6, 7, 9: Include in the legend the C/N values are reversed and why.
Figure S1 legend: Include that Enderby Island is part of the Auckland Islands archipelago, otherwise it
is a bit confusing as to why data from here are used for those that do not know this.

**Reviewer #2- Claudia Aracena:** Your manuscript "Late Holocene intensification of the westerly winds at the
1315 subantarctic Auckland Islands (51°S), New Zealand" is a very interesting contribution for the knowledge of past climate,
and ocean reconstructions at the Southern Hemisphere. In general, the quality of the information described in the
manuscript is good, as well as the figures and tables. We thank Claudia for these supportive comments. With humility
and to enhance the writing of your scientific contribution my main suggestions are: 1) to include in the study area
climatological modern data description related to precipitation and runoff discharge if exists (see Fig S1. Unfortunately,
1320 very little modern climatological data exists for the Auckland Islands so we are not able to implement this very good
suggestion.) and 2) re-organize the discussion, and results chapter. The discussion is too long and not focused on the
topic described in the title of this manuscript (We agree, and the following modifications have been made to the
discussion section: 1) LIA discussion has been cut from section 5.5 and moved to the end of section 5.4.1; 2) Additional
records of late Holocene SHWW variability from 51-53°S and the WAIS Divide ice core have been included to highlight
1325 late Holocene intensification of the westerlies related to atmospheric teleconnections of these areas to the tropical
Pacific 3). I believe that the chapters 5.4.1, 5.4.2, and 5.4.3 could be nicely summarized in a discussion table. Rather
than producing a table, we have streamlined this text and modified Fig. 9 to highlight key records. We believe these
changes address the reviewer's core concern.

1330 Also there are minor changes to be done in order to highlight the results of this interesting investigation.

1. Between lines 15 and 25: You mention "Drainage basin response","Hydrographic response", and "vegetation
response". All in relation to SHWW variability. I believe that is necessary to clarify such "responses" because is too
vague. This would make easier for the reader to have a better comprehension of what you want to say when you
1335 mention such "responses". We have modified the abstract text to clarify our intent.
2. There is an inconsistency with the term between the meaning of C/N lines 18, and 83. . . ."monitor influxes of
lithogenous, and terrestrial vs marine organic matter". . .Lithogenous has nothing to do with organic matter provenance
I believe. A comma has been added after 'magnetic susceptibility (MS) (Line 17) to make this sentence clearer. Now
this sentence should indicate that MS monitors influx of lithogenous material, whereas bulk organic $\delta^{13}$C and atomic
1340 C/N monitor the relative contribution of $OM_{terr}$ and $OM_{mar}$.
3. Between lines 87, and 94: I suggest addressing these questions to the discussion chapter. In the introduction, I
would only leave open the question that can be fully addressed by the results presented in this manuscript. We prefer
to leave these directed questions at the end of the introduction so the reader has a clear idea of the layout of the study
as presented in the methods, results, and discussion sections. We believe our manuscript addresses all three questions
1345 and further observe that this is a stylistic choice.
4. First paragraph between lines 117 to 121: I suggest to move this historical information to the introduction chapter.
We agree that Claudia's suggestion improves the flow of the manuscript. The following sentence has been moved and
*modified* from lines 117-121 to the second paragraph of the introduction; "Although the STF southwest of New Zealand
shifted rapidly south towards the *subantarctic* during the Last Glacial Maximum and deglaciation in response to an
1350 intensification/southward shift in SHWW, the position of this front has remained relatively stable over the Holocene
(Bostock et al., 2015), likely due to topographic steering at the 500 m depth contour (Smith et al., 2013)". The sentence
after that has also been modified; "The efficiency of the SAZ carbon sink *is sensitive to not only latitudinal migrations
of winds and associated fronts on glacial-interglacial timescales, but also to* decadal changes in atmospheric circulation
patterns, particularly, the symmetry of low-level zonal winds (Le Quéré et al., 2007; Metzl, 2009; Takahashi et al., 2009;
1355 Landschutzer et al. 2015)".
5. Between lines 121 to 125: I suggest to move this paragraph to continue in line 103 as. . .. At the present, shelf waters
surrounding the Auckland…. We elect to keep these sentences as a separate paragraph in the study area section as
they present the oceanographic setting as it pertains to the study area. Paragraphs 1 and 2 address physical geology
and glacial history, and climatological data respectively.
1360 6. Paragraph starting in line 123 to line 127: I suggest to move to the introduction chapter, considered as an author's
hypothesis that also justifies this study. We agree that this information would fit well in the introduction section and will

consider adding an additional sentence in the introduction to describe what we would expect the proxies from the sediment core to show. For the revised submission, however, we prefer to introduce our conceptual model immediately following the text describing the oceanography of the southern Campbell Plateau (Study Area), which provides important context. If the conceptual model were moved to the introduction, the reader would lack the necessary context to follow our argument and may be confused (eg. the nature of surrounding water masses is first mentioned in the Study Area section).

7. In the Methods chapter I suggest to move lines 137 and 140 (about seismic profiles results) to the result chapter of the manuscript. Following Claudia's suggestion, text from lines 137-140 has been removed from the methods section and placed at the beginning of the results section under a new heading '4.1 Seismic profiles'.

8. In the discussion chapter I suggest to move lines from 284 to 294 somewhere to the introduction chapter or include into the study area as a regional setting. As above, lines 284-294 have been removed from the discussion and added to the Introduction.

9. Between lines 299 to 300 you mention "we speculate". Also use this starting in sentence in line 318. I suggest for the discussion chapter starting sentences that enhance the value of the data presented in this investigation. We have removed the first use of "we speculate", which was unnecessary based on our observations of fjord hydrography. In the second instance, we are drawing a speculative link about a process for which supporting data are not yet available.

10. Error in line 471 because the cited reference does not describe inorganic carbon, it describes biogenic carbonate. 'Inorganic carbon' has been changed to biogenic carbonate.

11. Figure 1. There is a universe of stars in the figure. In general, I suggest resizing (to observe detailed view of piston corer as you wrote in the captions), and labelling again the panels because there are two figures more, not only one. Is necessary to indicate site 36P4 in bigger panel ( c). Would be interesting to include modern climatological data in the area (for example, precipitation curves and/or runoff discharge) since the interpretation of sediments is based on organic matter provenance proxies. See Fig. S1. I suggest to use another forms and colors to make differences on what is written in the text. See other error on figure: rhombus overlapping red star in the bigger ( c) panel. We have modified Figure 1 to label panel (d) (the detail map), and CTDs locations are now shown as circles. The caption has also been updated and we agree that this change has made things clearer (e.g. all stars in panels (a) and (b) refer to the same location). This change will also reduce the confusion introduced by plotting rhombuses on top of stars, although it is necessary to overlap symbols where both a piston core and CTD were taken at the same location. Finally, the caption has been updated to clearly state that core 36P4 was collected at Site 36 and core 39P4 was collected at site 39. We prefer not to change the labels on the figure because it is important to maintain the link between CTDs and cores collected from the same site.

12. Figure 2. Is necessary to indicate site on figure sites 36P4 and 39P4. See above.

13. Figure 7. In relation to interpretation on organic matter provenance I suggest to look at the paper of Perdue and Koprivnjak, 2007 where they use molar N to C ratios to define end members and discriminate between terrestrial and marine organic carbon. We have plotted N/C ratios for the bulk organic stratigraphy and our interpretation of the trends shown in our C/N ratio data does not change. Because our inshore water column particulate measurements likely contain terrestrial organic carbon, we lack the C and N concentration (and isotope) data to fully characterize a marine end member. For this reason, we have not attempted to quantify the relative proportion of marine and terrestrial organic material. However, this would be a useful metric and we will attempt to collect appropriate samples in the future.

I hope all these recommendations would improve the communication of your manuscript.
My best regards and success,
Claudia M. Aracena P.

**Page 13: [1] Deleted**                 **Microsoft Office User**                 **6/16/17 12:20:00 PM**

The current mutliproxy westerly wind reconstruction from Hanfield Inlet, and tree-stump reconstructions (Turney et al., 2016) indicate weakened westerly winds over the Auckland Islands from 1,000-900 yr BP, while winds remained strong over Fiordland, Southern New Zealand, until weakening at 750 yr BP (Knudson et al., 2011). Taken together, these results indicate a northward shift or contraction of the SHWW over New Zealand from 900-750 yr BP, roughly coincident with the beginning of the LIA, which is a period of cool summer temperatures (anomaly of -0.56 °C ±0.29 °C; Cook et al., 2002; Lorrey et al., 2014) and glacial advance in southern New Zealand (Lorrey et al., 2014; Putnam et al., 2010). Decreased wind strength recorded in Fiordland after 750 kyr that coincides with glacier advance in the Southern Alps can be reconciled if cooler summer temperatures are dominating glacier mass balance over precipitation associated with the SHWW. In addition, data reanalysis of precipitation patterns in Fiordland and the Southern Alps demonstrate that these areas are affected to different degrees by variations in the SAM and ENSO, also explaining regional differences in these records during the LIA (Ummenhofer et al., 2009). [Office1]

**Page 15: [2] Deleted**                 **Microsoft Office User**                 **6/16/17 12:22:00 PM**

Reconstructions of the SAM using south Pacific tree-ring records (Villalba et al., 1997, 2012) and the James Ross Island ice core (eastern Antarctic Peninsula; Abram et al., 2014) indicate a predominately negative phase of the SAM during the LIA. Some records from Antarctica also document the LIA, which is manifested as cool conditions in the Ross Sea (Bertler et al., 2011; Rhodes et al., 2012), and at Siple Dome (Mosley-Thompson et al., 1990), and periods of glacial advance along the western Antarctic Peninsula (Christ et al., 2015; Domack et al., 2001; Reilly et al., 2016; Shevenell et al., 1996), which is generally interpreted as an equatorial shift in the westerlies (negative phase of SAM). Across the Pacific Ocean in Patagonia, Moy et al., (2008) present a high resolution record that spans the LIA, and identifies the period from 400-150 yr BP as an intensification of wind at 51° S, and although our record does not cover this interval, this comparison to Fiordland suggests a breakdown in zonal symmetry across the Pacific during the LIA. Overall, SHWW behaviour during the LIA appears to be generally characterised by a northward shift, associated with a negative phase of the SAM. The regional response of climate in southern New Zealand to variations in both the SAM and ENSO may be contributing to local differences. In addition, comparison of SHWW reconstructions from New Zealand to those in southern South America and the western Antarctic Peninsula suggest that zonal symmetry in the winds across the Pacific breaks down during the LIA, although further high resolution records of wind strength with robust chronologies are required to fully assess this. Zonal symmetry of the winds during the Holocene is important to understand because we know that the position of the winds on longer timescales can affect global carbon cycling, where a northward (southward) position of the SHWW is generally associated with decreased (increased) atmospheric $CO_2$ (Anderson et al., 2009; Sigman et al., 2010; Skinner et al., 2010; Toggweiler et al., 2006).

---

## Author Response (AR2)

**Title:** Late Holocene intensification of the westerly winds at the subantarctic Auckland Islands (51ºS), New Zealand

**Authors:** Imogen Browne, Christopher Moy, Christina Riesselman, Helen Neil, Lorelei Curtin, Andrew Gorman, and Gary Wilson

Revision notes in response to editor's comments are provided in blue text.

Why do the Temperature and salinity and other variables don't have error bars? The temperature and salinity profiles do not have error bars, because these variables are continuously measured in the water column by the CTD sensors that have accuracies of ± 0.001 °C and ± 0.0003 S/m, which is 2-3 orders of magnitude smaller than the changes we observe in the water column. We have updated the error bars for all measured geochemical data ($\delta^{18}$O, $\delta^{13}$C, $\delta^{15}$N, and C/N) that were collected during these CTD casts and have used the average standard deviation of replicate analyses for error bar width. We overlooked the fact that some of our measurements lacked error bars and we are grateful for the opportunity to correct this before publication.

What exactly does "dry" and "rainy" conditions mean? I would like to understand if these profiles are really representative of "dry" and "wet" conditions…. is it just 1 profile or an average of samples from various days?

CTD006 (Fig. 4b) is a single profile collected immediately after a low-pressure system passed over the Auckland Islands and produced strong W/SW winds, precipitation, and cool air temperatures. These weather conditions mixed the fjord water column and produced the uniform temperature, salinity and geochemical profiles in the water column seen in this CTD cast.

CTD007 (Fig. 4a) is a single profile collected after high pressure was building over the Auckland Islands for 48 hours. Calm winds, no precipitation, relatively warm temperatures, and limited cloud cover (sunny conditions) during this time caused the upper part of the water column to warm quickly and cause weak stratification in the upper 6-8 m of the water column.

Taken together, we use these two CTD casts as synoptic end members of how the westerly winds would influence the fjord hydrography over longer timescales (seasonal to centennial). If the winds over the Auckland Islands were weak for a sustained amount of time, we would expect the water column to have some degree of thermal and chemical stratification (like CTD007). Conversely, sustained strong westerly winds would produce prolonged conditions in the fjord like CTD006.

We have updated the main text (lines 349-352) and the figure caption (lines 1289-1292) to clarify these conditions.

In the text the reference to the "Landschützer" paper are misspelled. This reference has been corrected in the text.